# Formulating Discrete Probability Flow Through Optimal Transport

**Pengze Zhang** *
Sun Yat-sen University
zhangpz3@mail2.edu.cn

**Hubery Yin**\*
WeChat, Tencent Inc.
hubery@tencent.com

**Chen Li**
WeChat, Tencent Inc.
chaselli@tencent.com

**Xiaohua Xie** †
Sun Yat-sen University
xiexiaoh6@mail.edu.cn

## Abstract

Continuous diffusion models are commonly acknowledged to display a deterministic probability flow, whereas discrete diffusion models do not. In this paper, we aim to establish the fundamental theory for the probability flow of discrete diffusion models. Specifically, we first prove that the continuous probability flow is the Monge optimal transport map under certain conditions, and also present an equivalent evidence for discrete cases. In view of these findings, we are then able to define the discrete probability flow in line with the principles of optimal transport. Finally, drawing upon our newly established definitions, we propose a novel sampling method that surpasses previous discrete diffusion models in its ability to generate more certain outcomes. Extensive experiments on the synthetic toy dataset and the CIFAR-10 dataset have validated the effectiveness of our proposed discrete probability flow. Code is released at: https://github.com/PangzeCheung/Discrete-Probability-Flow.

## 1 Introduction

The emerging diffusion-based models [43, 20, 45, 46] have been proven to be an effective technique for modeling data distribution, and generating high-quality texts [31, 14], images [34, 11, 40, 37, 38, 21] and videos [22, 19, 39, 51, 17]. Considering their generative capabilities have surpassed the previous state-of-the-art results achieved by generative adversarial networks [11], there has been a growing interest in exploring the potential of diffusion models in various advanced applications [41, 33, 48, 55, 10, 32, 49, 52, 18, 53].

Diffusion models are widely recognized for generating samples in a stochastic manner [46], which complicates the task of defining an encoder that translates a sample to a certain latent space. For instance, by following the configuration proposed by [20], it has been observed that generated samples from any given initial point have the potential to span the entire support of the data distribution. To achieve a deterministic sampling process while preserving the generative capability, Song *et al.*[46] proposed the probability flow, which provides a deterministic map between the data space and the latent space for continuous diffusion models. Unfortunately, the situation differs when it comes to discrete models. For instance, considering two binary distributions $(P_0 = \frac{1}{2}, P_1 = \frac{1}{2})$ and $(P_0 = \frac{1}{3}, P_1 = \frac{2}{3})$, there is no deterministic map that can transform the former distribution to the latter one, as it would simply be a permutation. Although some previous research has been

---

*Equal contribution. This work was done when Pengze Zhang was an intern at WeChat.
†Corresponding author.

37th Conference on Neural Information Processing Systems (NeurIPS 2023).

conducted on discrete diffusion models with discrete [24, 23, 4, 12, 9, 26, 16] and continuous [6, 47] time configurations, these works primarily focus on improving the sampling quality and efficiency, while sampling certainty has received less attention. More specifically, there is a conspicuous absence of existing literature addressing the probability flow in discrete diffusion models.

The aim of this study is to establish the fundamental theory of the probability flow for discrete diffusion models. Our paper contributes in the following ways. Firstly, we provide proof that under some conditions the probability flow of continuous diffusion coincides with the Monge optimal transport map during any finite time interval within the range of $(0, \infty)$. Secondly, we propose a discrete analogue of the probability flow under the framework of optimal transport, which we have defined as the *discrete probability flow*. Additionally, we identify several properties that are shared by both the continuous and discrete probability flow. Lastly, we propose a novel sampling method based on the aforementioned observations, and we demonstrate its effectiveness in significantly improving the certainty of the sampling outcomes on both synthetic toy dataset and CIFAR-10 dataset.

Proofs for all Propositions are given in the Appendix. For consistency, the probability flow and infinitesimal transport of a process $X_t$ is signified by $\hat{X}_t$ and $\tilde{X}_t$ respectively.

## 2 Background on Diffusion Models and Optimal Transport

First of all, we review some important concepts from the theory of diffusion models, optimal transport and gradient flow.

### 2.1 Continuous state diffusion models

Diffusion models are generative models that consist of a forward process and a backward process. The forward process transforms the data distribution $p_{data}(x_0)$ into a tractable reference distribution $p_T(x_T)$. The backward process then generates samples from the initial points drawn from $p_T(x_T)$. According to [28], the forward process is modeled as the (time-dependent) Ornstein-Uhlenbeck (OU) process:

$$dX_t = -\theta_t X_t dt + \sigma_t dB_t, \tag{1}$$

where $\theta_t \geq 0, \sigma_t > 0, \forall t \geq 0$ and $B_t$ is the Brownian Motion (BM). The backward process is the reverse-time process of the forward process [2]:

$$dX_t = [-\theta_t X_t - \sigma_t^2 \nabla_{X_t} \log p(X_t, t)]dt + \sigma_t d\tilde{B}_t, \tag{2}$$

where $\tilde{B}_t$ is the reverse-time Brownian motion and $p(X_t, t)$ is the single-time marginal distribution of the forward process, which also serves as the solution to the Fokker-Planck equation [35]:

$$\frac{\partial}{\partial t} p(x, t) = \theta_t \nabla_x(x p(x, t)) + \frac{1}{2}\sigma_t^2 \Delta_x p(x, t). \tag{3}$$

In order to train a diffusion model, the primary objective is to minimize the discrepancy between the model output $s_\theta(x_t, t)$ and the Stein score function $s(x_t, t) = \nabla_{x_t} \log p(x_t, t)$ [25]. Song *et al.* [45] demonstrate that, it is equivalent to match $s_\theta(x_t, t)$ with the conditional score function:

$$\theta^* = \arg\min_\theta \mathbb{E}_t \left\{ \lambda_t \mathbb{E}_{x_0, x_t} \left[ \|s_\theta(x_t, t) - \nabla_{x_t} \log p(x_t, t|x_0, 0)\|^2 \right] \right\}, \tag{4}$$

where $\lambda_t$ is a weighting function, $t$ is uniformly sampled over $[0, T]$ and $p(x_t, t|x_0, 0)$ is the forward conditional distribution.

It is noted that every Ornstein-Uhlenbeck process has an associated probability flow, which is a deterministic process that shares the same single-time marginal distribution [46]. The probability flow is governed by the following Ordinary Differential Equation (ODE):

$$d\hat{X}_t = [-\theta_t \hat{X}_t - \frac{1}{2}\sigma_t^2 s(\hat{X}_t, t)]dt. \tag{5}$$

In accordance with the global version of Picard-Lindelöf theorem [1] and the adjoint method[36, 7], the map

$$\begin{aligned} T_{s,t} : \mathbb{R}^n &\longrightarrow \mathbb{R}^n, \\ \hat{X}_s &\longmapsto \hat{X}_t. \end{aligned} \tag{6}$$

is a diffeomorphism $\forall t \geq s > 0$. The diffeomorphism naturally gives a transport map.

## 2.2 Discrete state diffusion models

In the realm of discrete state diffusion models, there are two primary classifications: the Discrete Time Discrete State (DTDS) models and the Continuous Time Discrete State (CTDS) models, which are founded on Discrete Time Markov Chains (DTMC) and Continuous Time Markov Chains (CTMC), correspondingly. Campbell *et al.*[6] conducted a comparative analysis of these models and determined that CTDS outperforms DTDS. The DTDS models construct the forward process through the utilization of the conditional distribution $q_{t+1|t}(x_{t+1}|x_t)$ and employ a neural network to approximate the reverse conditional distribution $q_{t|t+1}(x_t|x_{t+1}) = \frac{q_{t+1|t}(x_{t+1}|x_t)q_t(x_t)}{q_{t+1}(x_{t+1})}$. In practical applications, it is preferable to parameterize this model using $p^\theta_{0|t+1}$ [24, 4] and obtain $p^\theta_{k|k+1}$ through

$$
\begin{aligned}
p^\theta_{k|k+1}(x_k|x_{k+1}) &= \sum_{x_0} q_{k|k+1,0}(x_k|x_{k+1}, x_0) p^\theta_{0|k+1}(x_0|x_{k+1}) \\
&= \sum_{x_0} q_{k+1|k}(x_{k+1}|x_k) \frac{q_{k|0}(x_k|x_0)}{q_{k+1|0}(x_{k+1}|x_0)} p^\theta_{0|k+1}(x_0|x_{k+1}).
\end{aligned}
\tag{7}
$$

In contrast to DTDS models, a CTDS model is characterized by the (infinitesimal) generator [3], or transition rate, $Q_t(x, y)$. The Kolmogorov forward equation [13] is:

$$
\frac{\partial}{\partial t} q_{t|s}(x_t|x_s) = \sum_y q_{t|s}(y|x_s) Q_t(y, x_t).
\tag{8}
$$

The reverse process is:

$$
\frac{\partial}{\partial s} q_{s|t}(x_s|x_t) = \sum_y q_{s|t}(y|x_t) R_t(y, x_s).
\tag{9}
$$

The generator of the reverse process can be written by [6, 47]:

$$
R_t(y, x) = \frac{q_t(x)}{q_t(y)} Q_t(x, y) = \sum_{y_0} \frac{q_{t|0}(x|y_0)}{q_{t|0}(y|y_0)} q_{0|t}(y_0|y) Q_t(x, y).
\tag{10}
$$

There are various approaches to train the model, such as the Evidence Lower Bound (ELBO) technique [6], and the score-based approach [47]. It has been observed that the reverse generator can be factorized over dimensions, allowing parallel sampling for each dimension during the reverse process. However, it is important to note that this independence is only possible when the time interval for each step is small.

## 2.3 Optimal transport

The *optimal transport problem* can be formulated in two primary ways, namely the Monge formulation and the Kantorovich formulation [42]. Suppose there are two probability measures $\mu$ and $\nu$ on $(\mathbb{R}^n, \mathcal{B})$, and a cost function $c : \mathbb{R}^n \times \mathbb{R}^n \to [0, +\infty]$. The *Monge problem* is

$$
\text{(MP)} \inf_{\text{T}} \left\{ \int c(x, \text{T}(x)) \, d\mu(x) : \text{T}_\# \mu = \nu \right\}.
\tag{11}
$$

The measure $\text{T}_\# \mu$ is defined through $\text{T}_\# \mu(A) = \mu(\text{T}^{-1}(A))$ for every $A \in \mathcal{B}$ and is called the *pushforward* of $\mu$ through T.

It is evident that the Monge Problem (MP) transports the entire mass from a particular point, denoted as $x$, to a single point $\text{T}(x)$. In contrast, Kantorovich provided a more general formulation, referred to as the *Kantorovich problem*:

$$
\text{(KP)} \inf_{\gamma} \left\{ \int_{\mathbb{R}^n \times \mathbb{R}^n} c \, d\gamma : \gamma \in \Pi(\mu, \nu) \right\},
\tag{12}
$$

where $\Pi(\mu, \nu)$ is the set of *transport plans*, i.e.,

$$
\Pi(\mu, \nu) = \left\{ \gamma \in \mathcal{P}(\mathbb{R}^n \times \mathbb{R}^n) : (\pi_x)_\# \gamma = \mu, (\pi_y)_\# \gamma = \nu \right\},
\tag{13}
$$

where $\pi_x$ and $\pi_y$ are the two projections of $\mathbb{R}^n \times \mathbb{R}^n$ onto $\mathbb{R}^n$. For measures absolutely continuous with respect to the Lebesgue measure, these two problems are equivalent [50]. However, when the measures are discrete, they are entirely distinct as the constraint of the Monge Problem may never be fulfilled.

## 2.4 Fokker-Planck equation by gradient flow

According to [27], the Fokker-Planck equation represents the gradient flow of a functional in a metric space. In particular, for Brownian motion, its Fokker-Planck equation, which is also known as the heat diffusion equation, can be expressed as:

$$\frac{\partial}{\partial t} p(x,t) = \frac{1}{2} \Delta p(x,t), \tag{14}$$

and it represents the gradient flow of the Gibbs-Boltzmann entropy multiplied by $-\frac{1}{2}$:

$$-\frac{1}{2} S(p) = \frac{1}{2} \int_{\mathbb{R}^n} p(x) \log p(x) \, \mathrm{d}x. \tag{15}$$

It is worth noting that Eq. 15 is the gradient flow of Eq. 14 under the *2-wasserstein metric* ($W_2$).

Chow *et al.* [8] have developed an analogue in the discrete setting by introducing the discrete Gibbs-Boltzmann entropy:

$$S(p) = \sum_i p_i \, log \, p_i, \tag{16}$$

and deriving the gradient flow using a newly defined metric (Definition 1 in [8]). Since the discrete model is defined on graph $G(V, E)$, where $V = \{a_1, ..., a_N\}$ is the set of vertices, and $E$ is the set of edges, the discrete Fokker-Planck equation with a constant potential can be written as:

$$\frac{d}{dt} p_i = \sum_{j \in N(i)} p_j - p_i, \tag{17}$$

where $N(i) = \{j \in \{1, 2, ..., N\} | \{a_i, a_j\} \in E\}$ represents the one-ring neighborhood.

# 3 Continuous probability flow

## 3.1 The equivalence of Ornstein-Uhlenbeck processes and Brownian motion

The diffusion models that are commonly utilized in machine learning are founded on Ornstein-Uhlenbeck processes. First of all, we demonstrate that it is feasible to deterministically convert a time-dependent Ornstein-Uhlenbeck process into a standard Brownian motion.

**Proposition 1.** *Let $X_t$ and $Y_t$ be a time-dependent Ornstein-Uhlenbeck process and a Brownian motion respectively: $dX_t = -\theta_t X_t dt + \sigma_t dB_t^{(1)}$, $dY_t = dB_t^{(2)}$, where $B_t^{(1)}$ and $B_t^{(2)}$ are two independent Brownian motions and $\theta_t \geq 0, \sigma_t > 0, \forall t \geq 0$. Let $\phi_t = \exp(\int_0^t \theta_\tau \, \mathrm{d}\tau)$, $\beta_t = \int_0^t (\sigma_\tau \phi_\tau)^2 \, \mathrm{d}\tau$. Then $X_t$ coincides in law with $\phi_t^{-1} Y_{\beta_t}$.*

Building upon the aforementioned proposition, the primary focus of this paper is centered around the standard Brownian motion $dY_t = dB_t$.

## 3.2 Probability flow is a Monge map

Khrulkov *et al.* [29] have proposed a conjecture that the probability flow of Ornstein-Uhlenbeck process is a Monge map. However, they only provided a proof for a simplified case. We demonstrate that under some conditions, the conjecture is correct.

It is important to highlight that the continuous optimal transports presented in this paper are defined exclusively with the cost function: $c(x, y) = \frac{1}{2} |x - y|^2$.

Within the context of generative models, a collection of training samples denoted as $\{x_i\}_{i=1}^N$ is typically provided, and these samples are intrinsically defined by a distribution:

$$p(x, 0) = \frac{1}{N} \sum_{i=1}^N \delta(x - x_i), \tag{18}$$

where $\delta(x)$ represents the Dirac delta function. Given a Brownian motion with an initial distribution in the form of Equation (18), the single-time marginal distribution is [35]

$$p_B(x,t) = \frac{1}{N}\sum_{i=1}^{N}(2\pi t)^{-\frac{n}{2}}\exp(-\frac{|x-x_i|^2}{2t}).$$ 
(19)

The probability flow is defined as [46]:

$$d\hat{Y}_t = -\frac{1}{2}\nabla_{\hat{Y}_t}\log p_B(\hat{Y}_t,t)dt.$$ 
(20)

According to [1, 36, 7], the solution exists for all $t > 0$ and the map $\hat{Y}_{t+s}(\hat{Y}_t)$ is a diffeomorphism for all $t > 0, s \geq 0$. We have discovered that $\hat{Y}_{t+s}(\hat{Y}_t)$ is the Monge map under some conditions and the time does not reach $0$ or $+\infty$.

**Proposition 2.** *Given that $Y_0$ follows the initial condition (18), and all $x_i$s lie on the same line, the diffeomorphism $\hat{Y}_{t+s}(\hat{Y}_t)$ is the Monge optimal transport map between $p_B(x,t)$ and $p_B(x,t+s)$, $\forall\, t > 0, s \geq 0$.*

There is a counterexample [30] to demonstrate that the probability flow map does not necessarily provide optimal transport. It is important to note that their case differs from our assumptions in two ways. Firstly, they consider the limit case of $\hat{Y}_{+\infty}(\hat{Y}_0)$. Secondly, the initial distribution of the counterexample does not conform to the form specified in Equation (18). Therefore, their counterexample is not applicable to our situation.

It has been shown that the heat diffusion equation can be regarded as the *gradient flow* of the Gibbs-Boltzmann entropy concerning the $W_2$ *metric* [27]. As $W_2$ is associated with optimal transport, it is reasonable to anticipate that the "infinitesimal transport" $\hat{Y}_{t+dt}(\hat{Y}_t)$ is optimal [29].

In order to interpret the concept of "infinitesimal transport", we utilize the generator of the process $Y_t$. Let $C_c^2(\mathbb{R}^n)$ denote the set of twice continuously differentiable functions on $\mathbb{R}^n$ with compact support. The generator $A_t$ is defined as follows [35]:

$$\hat{A}_t f = \lim_{\Delta t \to 0^+}\frac{f(\hat{Y}_{t+\Delta t}) - f(\hat{Y}_t)}{\Delta t}, \forall f \in C_c^2(\mathbb{R}^n).$$ 
(21)

It is straightforward to verify that

$$\hat{A}_t = -\frac{1}{2}\nabla_x \log p_B(x,t)^T \nabla_x.$$ 
(22)

We define the "infinitesimal transport" to be the diffeomorphism $\tilde{Y}_{t+s}(\tilde{Y}_t)$ where $\tilde{Y}_{t+s}$ evolves according to the following equation

$$d\tilde{Y}_{t+s} = -\frac{1}{2}\nabla_{\tilde{Y}_t}\log p_B(\tilde{Y}_t(\tilde{Y}_{t+s}),t)ds,$$ 
(23)

with the initial condition $\tilde{Y}_t = \hat{Y}_t$. The generator of $\tilde{Y}_{t+s}$ is

$$\tilde{A}_{t+s} = -\frac{1}{2}\nabla_{\tilde{Y}_t}\log p_B(\tilde{Y}_t(\tilde{Y}_{t+s}),t)\nabla_x.$$ 
(24)

**Proposition 3.** *Given any $t > 0$, there exists a $\delta_t > 0$ s.t. $\forall\, 0 < s < \delta_t$, the diffeomorphism $\tilde{Y}_{t+s}(\tilde{Y}_t)$ with the initial condition $\tilde{Y}_t = \hat{Y}_t$ is the Monge optimal transport map.*

Let us return to the original Ornstein-Uhlenbeck process $X_t$. As it is merely a deterministic transformation of the Brownian motion $Y_t$, we can anticipate that the probability flow of $X_t$, denoted by $\hat{X}_t$, will be a Monge map. In fact, this expectation holds true:

**Proposition 4.** *Given that $X_0$ follows the initial condition (18), and all $x_i$s lie on the same line, the diffeomorphism $\hat{X}_{t+s}(\hat{X}_t)$ is the Monge optimal transport map for all $t > 0, s \geq 0$.*

# 4 Discrete probability flow

The continuous probability flow is deterministic, which means the "mass" at $\hat{Y}_t$ is entirely transported to $\hat{Y}_{t+s}$ during the time interval $[t, t+s]$. However, it is widely acknowledged that for discrete distributions $\mu$ and $\nu$, there may not exist a T such that $T_{\#}\mu = \nu$. As a result, discrete diffusions cannot possess a deterministic probability flow. To establish the concept of the *discrete probability flow*, we employ the methodology of optimal transport. First of all, a discrete diffusion model is proposed as an analogue of Brownian motion. Secondly, we modified the forward process to create an optimal transport map, which is used to define the discrete probability flow. Finally, a novel sampling technique is introduced, which significantly improves the certainty of the sampling outcomes.

## 4.1 Constructing discrete probability flow

It is demonstrated that the process described by Equation (17) is a discrete equivalent of the heat diffusion process (14) [8]. We adopt this process as our discrete diffusion model and represent it in a more comprehensive notation.

The discrete diffusion model has $K$ dimensions and $S$ states. The states are denoted by $i = (i_1, i_2, \ldots, i_K)$, where $i_j \in \{1, 2, \ldots, S\}$. The Kolmogorov forward equation for this process is

$$\frac{d}{dt} P_j^i(t|s) = \sum_{j'} P_{j'}^i(t|s) Q_{D_j^{j'}}(t), \tag{25}$$

where $P_j^i(t|s)$ means $P(x_t = j | x_s = i)$ and $Q_D$ is defined as:

$$Q_{D_j^i} = \begin{cases} 1, & d_D(i,j) = 1, \\ -\sum_{j' \in \{k:d_D(i,k)=1\}} Q_{D_{j'}^i}, & d_D(i,j) = 0, \\ 0, & otherwise, \end{cases} \tag{26}$$

where $d_D(i,j) = \sum_{l=1}^K |i_l - j_l|$. If we let the solution of the Equation (25) be denoted by $P_D(t|s)$ and assume an initial condition $P_0$, the single-time marginal distribution can be computed as follows:

$$P_{D_i}(t) = \sum_j P_{0j} Q_{D_i^j}(t|0). \tag{27}$$

It is noteworthy that the process defined by $Q_D$ is not an optimal transport map, as there exist *mutual flows* between the states (i.e., there exists two states $i, j$ with $Q_j^i > 0$ and $Q_i^j > 0$). Therefore, we propose a modified version that will be proved to be a solution to the Kantorovich problem, namely, an optimal transport plan. The modified version is defined by the following $Q$:

$$Q_j^i(t) = \begin{cases} \frac{ReLU(P_{D_i}(t) - P_{D_j}(t))}{P_{D_i}(t)}, & d_D(i,j) = 1, \\ -\sum_{j' \in \{k:d_D(i,k)=1\}} Q_{j'}^i(t), & d_D(i,j) = 0, \\ 0, & otherwise. \end{cases} \tag{28}$$

where

$$ReLU(x) = \begin{cases} x, & x > 0, \\ 0, & x \le 0. \end{cases} \tag{29}$$

In order to avoid singular cases, We define $Q_j^i(t)$ to be 0 when $P_{D_i}(t) = 0$. In fact, it is easy to verify that $P_{D_i}(t) > 0$ for all $t > 0$, $i \in \{1, 2, \ldots, K\}$. We will show that the process defined by $Q$ is equivalent in distribution to the one generated by $Q_D$.

**Proposition 5.** *The processes generated by $Q_D$ and $Q$ have the same single-time marginal distribution $\forall t > 0$.*

**Proposition 6.** *Given any $t > 0$, there exists a $\delta_t > 0$ s.t. $\forall\, 0 < s < \delta_t$, the process generated by $Q$ provides an optimal transport map from $P_D(t)$ to $P_D(t+s)$ under the cost $d_D$.*

Proposition 6 demonstrates that $Q_D$ generates a Kantorovich plan between $P_D(t)$ and $P_D(t+s)$ under a certain cost function. On the other hand, the continuous probability flow is the Monge map between $p_B(x,t)$ and $p_B(x,t+s)$. Therefore, it is reasonable to define the process defined by $Q_D$ as the *discrete probability flow* of the original process defined by $Q$.

Furthermore, the "infinitesimal transport" of the discrete process, which is defined by $\frac{d}{ds}\hat{P}(t+s) = \hat{P}(t+s)Q(t)$, also provides an optimal transport map.

**Proposition 7.** *Given any $t > 0$, there exists a $\delta_t > 0$ s.t. $\forall 0 < s < \delta_t$, the process above provides an optimal transport map from $\hat{P}(t)$ to $\hat{P}(t+s)$ under the cost $d_D$.*

### 4.2 Sampling by discrete probability flow

In order to train the modified model, we employ a score-based method described in the Score-based Continuous-time Discrete Diffusion Model (SDDM) [47]. Specifically, we directly learn the conditional probability $P^\theta(i_l(t)|\{i_1,\ldots,i_{l-1},i_{l+1},\ldots,i_K\}(t))$. According to proposition 5, it follows that $P^\theta = P^\theta{}_D$, and consequently, the training process is identical to that of [47]. For the sake of brevity, we will employ the notation $P^\theta_{i_l|i\backslash i_l}(t)$ to replace $P^\theta(i_l(t)|\{i_1,\ldots,i_{l-1},i_{l+1},\ldots,i_K\}(t))$.

The generator of the reverse process is

$$R^i_j(t) = \begin{cases} ReLU(\frac{P^\theta_{D_{j_l|i\backslash i_l}}(t)}{P^\theta_{D_{i_l|i\backslash i_l}}(t)} - 1), & d_D(i,j) = 1 \text{ and } i_l \neq j_l, \\ -\sum_{j' \in \{k:d_D(i,k)=1\}} R^i_{j'}(t), & d_D(i,j) = 0, \\ 0, & otherwise. \end{cases} \tag{30}$$

We use the Euler's method to generate samples. Given the time step length $\epsilon$, the transition probabilities for dimension $l$ is:

$$P^\theta(i_l(t-\epsilon)|i(t)) = \begin{cases} \epsilon R^{i(t)}_{i_1(t),\ldots,i_l(t-\epsilon),\ldots,i_k(t)}(t), & i_l(t-\epsilon) \neq i_l(t), \\ 1 + \epsilon R^{i(t)}_{i(t)}(t), & i_l(t-\epsilon) = i_l(t). \end{cases} \tag{31}$$

When $\epsilon$ is small, the reverse conditional distribution has the factorized probability:

$$P^\theta(i(t-\epsilon)|i(t)) = \Pi^K_{l=1} P^\theta(i_l(t-\epsilon)|i(t)) \tag{32}$$

In this way, it becomes possible to generate samples by sequentially sampling from the reverse conditional distribution 32.

**Transition to higher probability states** The reverse process of the continuous probability flow, as described in Equation (20), causes particles to move towards areas with higher logarithmic probability densities. As the logarithm function is monotonically increasing, this reverse flow pushes particles to higher probability density states. This phenomenon is also observed in the discrete probability flow. By examining the reverse generator, as shown in Equation (30), it can be determined that the transition rate $R^i_j(t) > 0$ only when the destination state $j$ has a higher probability than the source state $i$. This implies that transitions only occur in higher probability states. In contrast, the original continuous reverse process (2) and the discrete reverse process from (10) allow any transitions.

**Reduction of Standard Deviation** We measure the certainty of the sampling method by the expectation of the Conditional Standard Deviation (CSD):

$$CSD_{s,t}(X) = \mathbb{E}_{X_t}[\text{Std}(X_s|X_t)], \tag{33}$$

where $\text{Std}(X_s|X_t) = \text{Var}^{\frac{1}{2}}(X_s|X_t) = \mathbb{E}^{\frac{1}{2}}_{X_s}[X_s - \mathbb{E}_{X_s}[X_s|X_t]|X_t]$. $CSD_{s,t}(X)$ is 0 when the process is deterministic, such as the continuous probability flow. In the discrete situation, there does not exist any deterministic map. However, our discrete probability flow significantly reduces $CSD_{s,t}(X)$. Table 2 presents numerical evidence of this phenomenon. Therefore, we posit that the discrete probability flow enhances the certainty of the sampling outcomes.

Table 1: Comparison of generation quality for SDDM and DPF, in terms of MMD with Laplace kernel using bandwith=0.1. Lower values indicate superior quality.

| | 2spirals | 8gaussians | checkerboard | circles | moons | pinwheel | swissroll |
|---|---|---|---|---|---|---|---|
| | discrete dimension = 32, state size = 2 | | | | | | |
| SDDM | 2.18e-06 | 4.28e-06 | 1.33e-06 | 6.22e-06 | 5.62e-06 | 2.10e-06 | 4.27e-06 |
| DPF (ours) | 1.89e-05 | 1.09e-05 | 2.22e-05 | 3.27e-05 | 2.42e-05 | 1.60e-05 | 2.18e-05 |
| | discrete dimension = 16, state size = 5 | | | | | | |
| SDDM | 2.06e-4 | 1.01e-4 | 2.43e-4 | 1.74e-4 | 2.20e-4 | 3.37e-4 | 1.43e-4 |
| DPF (ours) | 3.87e-4 | 5.87e-4 | 4.93e-4 | 3.83e-4 | 3.43e-4 | 6.64e-4 | 3.20e-4 |
| | discrete dimension = 12, state size = 10 | | | | | | |
| SDDM | 5.52e-4 | 3.01e-4 | 4.39e-4 | 4.22e-4 | 2.71e-4 | 2.90e-4 | 3.39e-4 |
| DPF (ours) | 7.19e-4 | 3.49e-4 | 5.99e-4 | 6.65e-4 | 4.34e-4 | 4.14e-4 | 5.17e-4 |

Table 2: Comparison of certainty for SDDM and DPF, in terms of $CSD$ on 4,000 initial points, each of which has 10 generated samples. Lower values indicate superior certainty.

| | 2spirals | 8gaussians | checkerboard | circles | moons | pinwheel | swissroll |
|---|---|---|---|---|---|---|---|
| | discrete dimension = 32, state size = 2 | | | | | | |
| SDDM | 14.3053 | 14.1882 | 14.7433 | 14.4327 | 14.1739 | 14.0450 | 14.0548 |
| DPF (ours) | 2.1719 | 1.7945 | 2.0693 | 1.7210 | 2.0573 | 2.1834 | 1.8892 |
| | discrete dimension = 16, state size = 5 | | | | | | |
| SDDM | 14.4645 | 14.6143 | 14.6963 | 14.4807 | 14.2397 | 14.2466 | 14.2659 |
| DPF (ours) | 1.9711 | 1.9367 | 1.4172 | 1.7185 | 1.7668 | 1.9633 | 1.6665 |
| | discrete dimension = 12, state size = 10 | | | | | | |
| SDDM | 12.8463 | 12.7933 | 13.0158 | 12.9232 | 12.6665 | 12.7634 | 12.7880 |
| DPF (ours) | 1.8123 | 1.3178 | 1.1348 | 1.4625 | 1.4859 | 1.8435 | 1.5227 |

## 5   Related Work

The concept of probability flow was initially introduced in [46] as a deterministic alternative to the Itô diffusion. In the work [44], they presented the Denoising Diffusion Implicit Model (DDIM) and demonstrated its equivalence to the probability flow. Subsequently, [29] investigated the relationship between the probability flow and optimal transport. They hypothesized that the probability flow could be considered a Monge optimal transition map and provided a proof for a specific case. Additionally, they conducted numerical experiments that supported their conjecture, showing negligible errors. However, [30] has discovered an initial distribution that renders probability flow not optimal.

The discrete diffusion models were first introduced by [43], who considered a binary model. Following the success of continuous diffusion models, discrete models have garnered more attention. The bulk of research on discrete models has focused primarily on the design of the forward process [24, 23, 4, 5, 26, 16, 9]. Continuous time discrete state models were introduced by [6] and subsequently developed by [47].

## 6   Experiments

We conduct numerical experiments using our novel sampling method by Discrete Probability Flow (DPF) on synthetic data. The primary goal is to demonstrate that our method can generate samples of comparable quality with higher certainty.

Experiments are conducted on synthetic data using the same setup as SDDM [47], with the exception that we replaced the generator $Q$ with Equation (26). In addition to the binary situation ($S = 2$) studied in [47], we also perform experiments on synthetic data with the state size $S$ set to 5 and 10. To evaluate the quality of the generated samples, we generated 40,000 / 4,000 samples for binary data / other type of data using SDDM and DPF, and measured the Maximum Mean Discrepancy (MMD) with the Laplace kernel [15]. The results are shown in Table 1. It can be seen that the MMD value obtained using DPF is slightly higher than that of SDDM, which may be attributed to the structure of the reverse generator 10. Specifically, DPF approximates an additional term, $Q_t(y, x)$, with the neural network, which potentially introduces additional errors to the sampling process, leading to a higher MMD value compared to SDDM. However, such difference is minimal and does not significantly impact the quality of the generated samples. As evident from the visualization of the

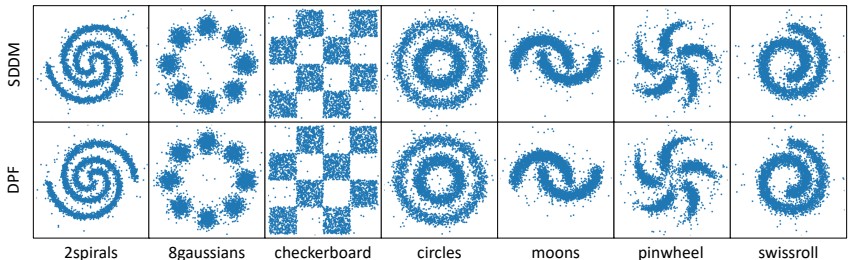

Figure 1: Visualization of the generation quality on generated binary samples for SDDM and DPF.

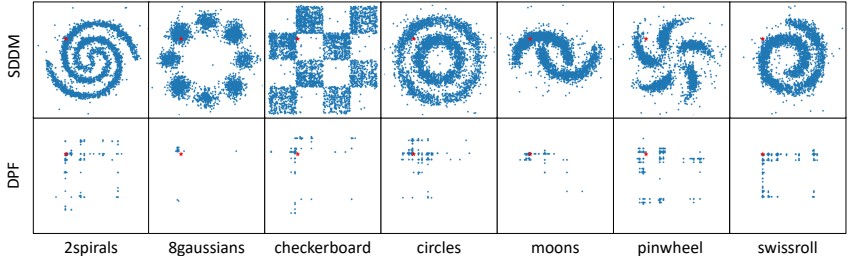

Figure 2: Visualization of the generating certainty on generated binary samples for SDDM and DPF. All the samples (in blue) are randomly generated from the single initial point (in red).

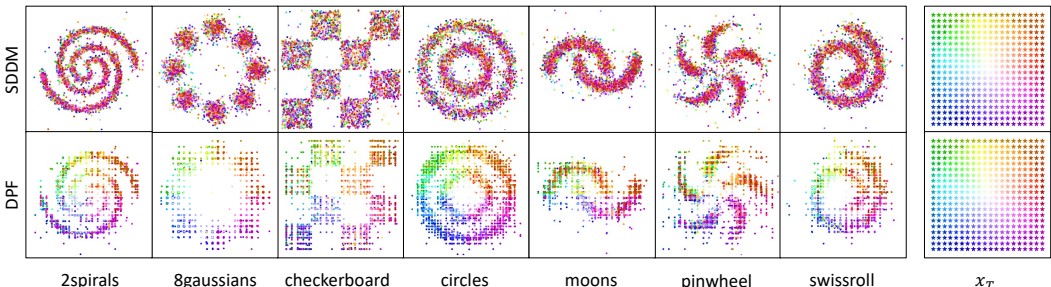

Figure 3: Visualization of the generated binary samples from the given initial points $\boldsymbol{x}_T$. Different colors distinguish the generated samples from different initial points $\boldsymbol{x}_T$.

distributions obtained from SDDM and DPF in Figure 1, it is clear that DPF can generate samples that are comparable to those generated by SDDM.

In addition, we also compare the sampling certainty of DPF and SDDM by computing $CSD_{s,t}$ using a Monte-Carlo based method. Specifically, we set $s = 0$ and $t = T$, and sample 4,000 $x_t$s with 10 $x_s$s for each $x_t$. We then estimate $\mathbb{E}(x_s|x_t)$ and $\text{Std}(x_s|x_t)$ using the sample mean and sample standard deviation, respectively. The results of certainty are presented in Table 2. Our findings indicate that DPF significantly reduces the $CSD$, which suggests a higher certainty. Additionally, we visualize the results of 4,000 generated samples (in blue) from a single initial point (in red) in the binary case in Figure 2. It is apparent that the sampling of SDDM exhibits high uncertainty, as it can sample the entire pattern from a single initial point. In contrast, our method reduces such uncertainty and is only able to sample a limited number of states.

To provide a more intuitive representation of the generated samples originating from various initial points, we select $20 \times 20$ initial points arranged in the grid, and distinguish them using different colors. Subsequently, we visualize the results by sampling 10 outcomes from each initial point, as shown in Figure 3. We observe that the visualization of SDDM samples appears disorganized, indicating significant uncertainty. In contrast, the visualization of DPF samples exhibits clear regularity, manifesting in two aspects: (1) the generated samples from the same initial point using DPF are clustered by color, demonstrating the better sampling certainty of our DPF. (2) Both of the generated samples and initial points are colored similarly at each position. For example, in the lower

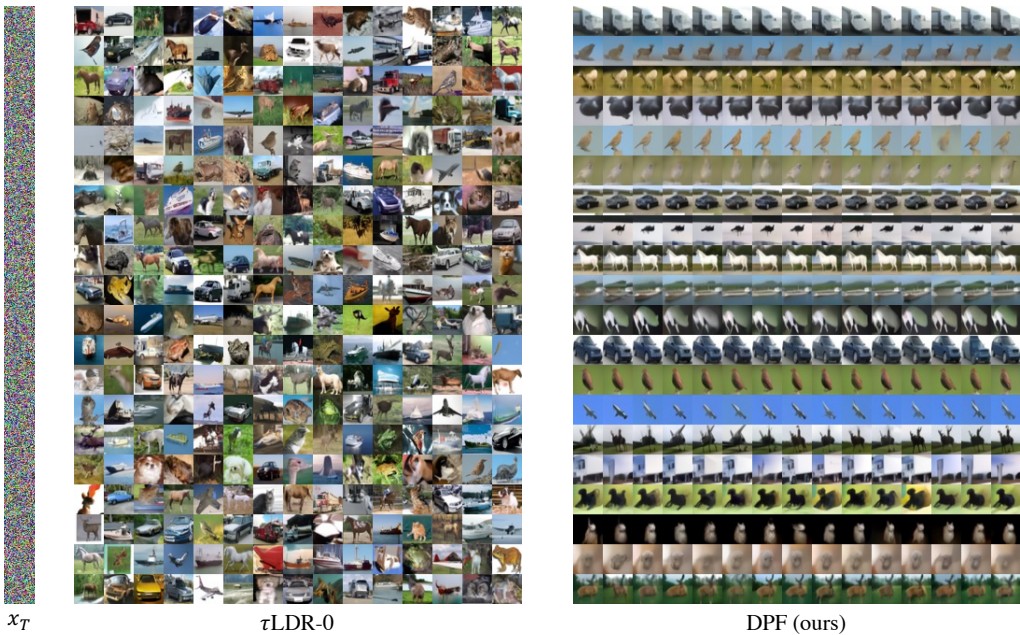

$x_T$                    $\tau$LDR-0                                    DPF (ours)

Figure 4: Image modeling on CIFAR-10 dataset. The figure is divided into three groups: initial points $x_T$, sampling results of $\tau$LDR-0, and sampling results of our DPF. For each row, the sampled images are obtained from the same initial point.

right area, a majority of the generated samples are colored purple, which corresponds to the color assigned to the initial points $x_T$ in that area. This observation demonstrates that most of the sampling results obtained through DPF are closer to their respective initial points, aligning with our design intention of optimal transport. It is worth noting that similar phenomena are observed across different state sizes, and we have provided these results in the Appendix.

Finally, we extended our DPF to the CIFAR-10 dataset, and compare it with the $\tau$LDR-0 method proposed in [6]. The visualization results are shown in Figure 4. It can be seen that our method greatly reduces the uncertainty of generating images by sampling from the same initial $x_T$. Detailed experimental settings and more experimental results are presented in the Appendix.

## 7   Discussion

In this study, we introduce a discrete counterpart of the probability flow and established its connections with the continuous formulations. We began by demonstrating that the continuous probability flow corresponds to a Monge optimal transport map. Subsequently, we proposed a method to modify a discrete diffusion model to achieve a Kantorovich plan, which naturally defines the discrete probability flow. We also discovered shared properties between continuous and discrete probability flows. Finally, we propose a novel sampling method that significantly reduces sampling uncertainty. However, there are still remaining aspects to be explored in the context of the discrete probability flow. For instance, to obtain more general conclusions under a general initial condition, the semi-group method [54] could be employed. Additionally, while we have proven the existence of a Kantorovich plan in a small time interval, it is possible to extend this to a global solution. Moreover, the definition of the probability flow has been limited to a specific type of discrete diffusion model, which also could be extended to a broader range of models. These topics remain open for future studies.

## 8   Acknowledgments and Disclosure of Funding

We would like to thank all the reviewers for their constructive comments. Our work was supported in National Natural Science Foundation of China (NSFC) under Grant No.U22A2095 and No.62072482.

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
