# Appendix

## Contents

## A   Overview of our DPF

To elucidate our methodology more intuitively, we include schematic diagrams in Figure 5, illustrating the sampling procedure from various diffusion models. Broadly speaking, diffusion models can be classified into two categories based on the nature of the underlying data space: continuous diffusion models and discrete diffusion models. Figure 5 (a) provides an illustration of a continuous diffusion model using a Stochastic Differential Equation (SDE) that transforms a prior noise distribution into

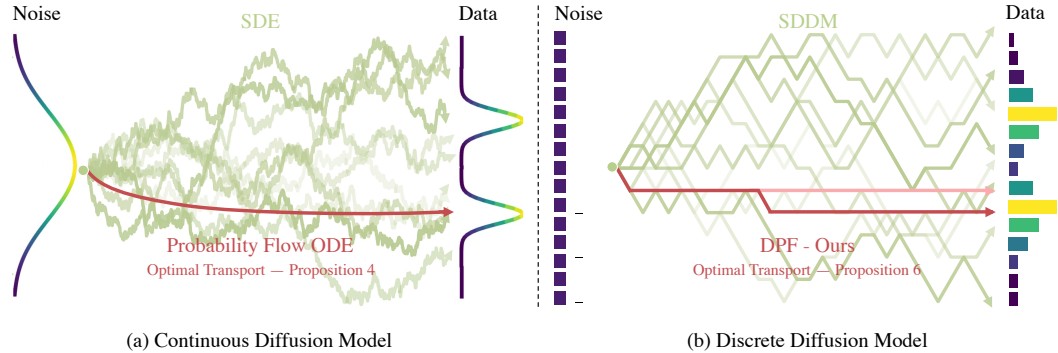

(a) Continuous Diffusion Model           (b) Discrete Diffusion Model

Figure 5: Schematic representation of different diffusion models.

the data distribution. The stochastic nature of the sampling process in continuous diffusion models allows samples generated from a single initial point to span the entire space (green line), but this feature limits its practical applicability. To overcome this limitation, probability flow is introduced, which ensures that the generated sample from an initial point follows a deterministic path (red line). This enhancement enables the continuous diffusion model to be more manageable and applicable in a broader range of scenarios.

In this paper, our concentration is primarily on discrete diffusion models. An example of such a model, based on SDDM with 15 states, is depicted in Figure 5 (b). Similar to SDE, it is observed that the sampling process is also susceptible to uncertainty (green line). One potential solution could involve incorporating probability flow into discrete diffusion in a similar manner as in the continuous models. Nonetheless, as previously mentioned in the introduction, this is not a viable option in discrete models due to the lack of a deterministic mapping between the latent space and the data space. Thus, there is a necessity for a redefined probability flow that is tailored to discrete diffusion models, and this forms the core of this paper. This study examines the probability flow of discrete diffusion models through the concept of optimal transport. Initially, we demonstrate that the continuous probability flow coincides with the Monge optimal transport map (Proposition 4). We then leverage this result to develop a similar probability flow for discrete diffusion models using optimal transport (Proposition 6). Finally, we propose a novel sampling methodology for discrete models that significantly reduces the uncertainty (red line) in the sampling process.

## B Definitions and Theorems Employed in this Appendix

For the sake of reader convenience, we hereby provide a comprehensive list of the definitions and theorems utilized in this paper. Additionally, we limit our representation to the case within $\mathbb{R}^n$.

**Theorem 1.** (Theorem 1.48 in [46]) *Suppose that $\mu$ is a probability measure on $(\mathbb{R}^n, \mathcal{B})$ such that $\int |x|^2 \, d\mu(x) < \infty$ and that $u : \mathbb{R}^n \to \mathbb{R} \cup \{+\infty\}$ is convex and differentiable $\mu$-a.e. Set $\mathrm{T} = \nabla u$ and suppose $\int |\mathrm{T}(x)|^2 \, d\mu(x) < \infty$. Then $\mathrm{T}$ is optimal for the transport cost $c(x, y) = \frac{1}{2}|x - y|^2$ between the measures $\mu$ and $\nu = \mathrm{T}_\# \mu$.*

**Definition 2.** *The optimization problem under constraint is formally defined as follows:*

$$\min_{x \in \mathbb{R}^n} f(x) \quad \text{subject to} \begin{cases} c_i(x) = 0, i \in \mathcal{E} \\ c_i(x) \geq 0, i \in \mathcal{I}. \end{cases} \tag{34}$$

*The Lagrangian for this constrained optimization problem is defined as:*

$$\mathcal{L}(x, \lambda) = f(x) - \sum_{i \in \mathcal{E} \cup \mathcal{I}} \lambda_i c_i(x). \tag{35}$$

*Here, $\lambda_i$ represents the Lagrange multiplier associated with the $i^{th}$ constraint. The active set at any feasible $x$ is defined as the union of the set $\mathcal{E}$ with the indices of the active inequality constraints, that is:*

$$\mathcal{A}(x) = \mathcal{E} \cup \{i \in \mathcal{I} : c_i(x) = 0\}. \tag{36}$$

**Definition 3.** (Definition 12.1 in [38]) *Given the point $x^*$, we say that the Linear Independence Constraint Qualification (LICQ) holds if the set of active constraint gradients $\{\nabla c_i(x^*), i \in \mathcal{A}(x^*)\}$ is linearly independent.*

**Theorem 4.** (Theorem 12.1 in [38], the Karush-Kuhn-Tucker (KKT) conditions) *Suppose that $x^*$ is a local solution of the problem (34) and that the LICQ holds at $x^*$. Then there is a Lagrange multiplier vector $\lambda^*$, with components $\lambda_i^*, i \in \mathcal{E} \cup \mathcal{I}$, such that the following conditions are satisfied at $(x^*, \lambda^*)$*

$$\nabla_x \mathcal{L}(x^*, \lambda^*) = 0, \tag{37a}$$
$$c_i(x^*) = 0, \quad \forall i \in \mathcal{E}, \tag{37b}$$
$$c_i(x^*) \geq 0, \quad \forall i \in \mathcal{I}, \tag{37c}$$
$$\lambda^* \geq 0, \quad \forall i \in \mathcal{I}, \tag{37d}$$
$$\lambda_i^* c_i(x^*) = 0, \quad \forall i \in \mathcal{E} \cup \mathcal{I}. \tag{37e}$$

*Remark 5.* According to Theorem 4, the Karush-Kuhn-Tucker (KKT) conditions serve as necessary conditions. In the case of linear programming, these conditions are not only necessary but also sufficient. To demonstrate this, let us consider the standard form of a linear programming problem:

$$\min c^T x, \quad \text{subject to } Ax = b, x \geq 0. \tag{38}$$

We can write the Lagrangian function for 38 as

$$\mathcal{L}(x, \pi, s) = c^T x - \pi^T (Ax - b) - s^T x. \tag{39}$$

The KKT conditions are

$$A^T \pi + s = c, \tag{40a}$$
$$Ax = b, \tag{40b}$$
$$x \geq 0, \tag{40c}$$
$$s \geq 0, \tag{40d}$$
$$x^T s = 0. \tag{40e}$$

Suppose we have a vector triple $(x^*, \pi^*, s^*)$ that satisfies Equation (40). In such a scenario, we can deduce that:

$$c^T x^* = (A^T \pi^* + s)^T x^* = (\pi^*)^T A x^* = b^T \pi^*. \tag{41}$$

Let us consider another feasible point denoted by $\bar{x}$, which satisfies the conditions $A\bar{x} = b$ and $\bar{x} \geq 0$. we can conclude that:

$$c^T \bar{x} = (A^T \pi^* + s^*)^T \bar{x} = b^T \pi^* + \bar{x}^T s^* \geq b^T \pi^* = c^T x^*. \tag{42}$$

The inequality (42) demonstrates that the KKT conditions serve as sufficient conditions.

**Theorem 6.** (Theorem 8.5.1 in [39]) *Let $X_t$ be an Itô diffusion given by*

$$\mathrm{d}X_t = b(X_t)\,\mathrm{d}t + \sigma(X_t)\,\mathrm{d}B_t, \quad b \in \mathbb{R}^n, \sigma \in \mathbb{R}^{n \times m}, X_0 = x, \tag{43}$$

*and let $Y_t$ be an Itô process given by*

$$\mathrm{d}Y_t = u(t, \omega)\,\mathrm{d}t + v(t, \omega)\,\mathrm{d}B_t, \quad u \in \mathbb{R}^n, v \in \mathbb{R}^{n \times m}, Y_0 = x. \tag{44}$$

*Assume that*

$$u(t, \omega) = c(t, \omega)b(Y_t) \quad and \quad vv^T(t, \omega) = c(t, \omega)\sigma\sigma^T(Y_t), \tag{45}$$

*for a.a. $t, \omega$. Define $\beta_t$ and $\alpha_t$ as:*

$$\beta_t = \beta(t, \omega) = \int_0^t c(s, \omega)\,\mathrm{d}s \quad and \quad \alpha_t = \inf\{s : \beta_s > t\}. \tag{46}$$

*Then $Y_{\alpha_t}$ coincides in law with $X_t$, denoted by $Y_{\alpha_t} \simeq X_t$.*

**Theorem 7.** (Theorem 4.1 of Chapter V, §4 in [32], Poincaré's lemma). *Let $U$ be an open ball in $\mathbb{R}^n$ and let $\omega$ be a differential form of degree $\geq 1$ on $U$ such that $\mathrm{d}\omega = 0$. Then there exists a differential form $\phi$ on $U$ such that $\mathrm{d}\phi = \omega$.*

*Remarks.* The conclusion remains valid when the open ball $U$ is substituted with the entirety of $\mathbb{R}^n$.

**Theorem 8.** (Theorem 4 in [5]) *The solution of the differential equation $Y' = A(t)Y$ with initial condition $Y(0) = Y_0$ can be written as $Y(t) = \exp(\Omega(t))Y_0$ with $\Omega(t)$ defined by*

$$\Omega' = \mathrm{d}\exp_\Omega^{-1}(A(t)), \quad \Omega(0) = O. \tag{47}$$

*where*

$$\mathrm{d}\exp_\Omega^{-1}(A(t)) = \sum_0^\infty \frac{B_k}{k!}\mathrm{ad}_\Omega^k(A), \tag{48}$$

*and $B_k$ is the Bernoulli numbers. $\mathrm{ad}_\Omega^k(A)$ is defined through*

$$\mathrm{ad}_\Omega(A) = [\Omega, A], \quad \mathrm{ad}_\Omega^j(A) = [\Omega, \mathrm{ad}_\Omega^{j-1}(A)], \quad \mathrm{ad}_\Omega^0(A) = A, \quad j \in \mathbb{N}, \tag{49}$$

*where $[A, B] = AB - BA$ is the Lie-bracket.*

*Remarks.* If $A(s)A(t) = A(t)A(s), \forall s, t \geq 0$, $\Omega(t)$ has the simple form $\Omega(t) = \int_0^t A(s)\,\mathrm{d}s$.

## C Proofs

### C.1 Proof of Proposition 1

*Proof.* By Itô formula:

$$\begin{aligned}
\mathrm{d}(\phi_t X_t) &= \phi_t \theta_t X_t \,\mathrm{d}t + \phi_t \,\mathrm{d}X_t \\
&= \phi_t \theta_t X_t \,\mathrm{d}t - \phi_t \theta_t X_t \,\mathrm{d}t + \phi_t \sigma_t \,\mathrm{d}B_t \\
&= \phi_t \sigma_t \,\mathrm{d}B_t.
\end{aligned} \tag{50}$$

By Theorem 6, $\phi_{\alpha_t} X_{\alpha_t} \simeq Y_t$, which means $X_t$ coincides in law with $\phi_t^{-1} Y_{\beta_t}$ $\qquad\square$

*Remarks.* Proposition 1 posits that the Ornstein-Uhlenbeck (OU) process is essentially a scaling of Brownian motion with a change in time. Consequently, the VE SDEs, VP SDEs, sub-VP SDEs in [50], as well as the models presented in [30], can be regarded as equivalent.

### C.2 Proof of Proposition 2

**Lemma B.2.1** *Let $H_t$ be the Hessian matrices $\nabla_{x_t}^2 \log p_B(x_t, t)$, then $H_s H_t = H_t H_s, \forall s, t \geq 0$.*

*Proof.*

$$\begin{aligned}
H_t &= \nabla_{x_t}^2 \log p_B(x_t, t) \\
&= \nabla_{x_t} \frac{\sum_i \exp(-\frac{|x_t - x_i|^2}{2t})(-\frac{x_t - x_i}{t})}{\sum_j \exp(-\frac{|x_t - x_j|^2}{2t})} \\
&= \underbrace{\sum_i \nabla_{x_t}\left(\frac{\exp(-\frac{|x_t - x_i|^2}{2t})}{\sum_j \exp(-\frac{|x_t - x_j|^2}{2t})}\right)\left(-\frac{x_t - x_i}{t}\right)}_{A} + \underbrace{\frac{\sum_i \exp(-\frac{|x_t - x_i|^2}{2t})}{\sum_j \exp(-\frac{|x_t - x_j|^2}{2t})}\left(-\frac{1}{t}\right)I}_{B}.
\end{aligned} \tag{51}$$

$B$ is a scalar matrix, then it commutes with any matrix.

$$
\begin{aligned}
A =& \underbrace{(\sum_j \exp(-\frac{|x_t - x_j|^2}{2t}))^{-2}}_{C} \sum_i [\exp(-\frac{|x_t - x_i|^2}{2t})(-\frac{x_t - x_i}{t}) \sum_j \exp(-\frac{|x_t - x_j|^2}{2t}) \\
& - \exp(-\frac{|x_t - x_i|^2}{2t}) \sum_j \exp(-\frac{|x_t - x_j|^2}{2t})(-\frac{x_t - x_j}{t})](-\frac{x_t - x_i}{t})^T \\
=& C \sum_{i,j} \exp(-\frac{|x_t - x_i|^2}{2t} - \frac{|x_t - x_j|^2}{2t})(\frac{x_t - x_i}{t})(\frac{x_t - x_i}{t})^T \\
& - C \sum_{i,j} \exp(-\frac{|x_t - x_i|^2}{2t} - \frac{|x_t - x_j|^2}{2t})(\frac{x_t - x_j}{t})(\frac{x_t - x_i}{t})^T \\
=& C \sum_{i<j} \exp(-\frac{|x_t - x_i|^2}{2t} - \frac{|x_t - x_j|^2}{2t})\frac{1}{t^2}[(x_t - x_i)(x_t - x_i) + (x_t - x_j)(x_t - x_j) \\
& - (x_t - x_j)(x_t - x_i)^T - (x_t - x_i)(x_t - x_j)^T] \\
=& C \sum_{i<j} \exp(-\frac{|x_t - x_i|^2}{2t} - \frac{|x_t - x_j|^2}{2t})\frac{1}{t^2}(x_j - x_i)(x_j - x_i)^T.
\end{aligned}
\tag{52}
$$

As $x_i$s lie on the same line, $x_j - x_i$ can be denoted by $x_j - x_i = C_{i,j}v, \forall i, j$, where $v$ is a fixed vector. It has $(x_j - x_i)(x_j - x_i)^T = C_{i,j}^2 vv^T$. It is clear that $C_{i,j}^2 vv^T$ and $C_{k,l}^2 vv^T$ commutes $\forall i, j, k, l$. Furthermore, as $t$ and $x_t$ only appear in the coefficients, $H_t$ and $H_s$ commute with one another. $\square$

*Proof of Proposition2.* If $Y_0$ follows the initial condition (18), and $x_i$s lie on the same line, $Y_t$ will governed by the equation (20). Employing the trick in [8], For a fixed $T$, we define

$$
a(t) = \nabla_{\hat{Y}_t} \hat{Y}_T.
\tag{53}
$$

Then we can derive

$$
\begin{aligned}
\frac{\mathrm{d}a(t)}{\mathrm{d}t} &= \lim_{\epsilon \to 0^+} \frac{a(t+\epsilon) - a(t)}{\epsilon} \\
&= \lim_{\epsilon \to 0^+} \frac{a(t+\epsilon) - a(t+\epsilon)\nabla_{\hat{Y}_t}(\hat{Y}_t - \epsilon\frac{1}{2}\nabla_{\hat{Y}_t} \log p_B(\hat{Y}_t, t) + \mathcal{O}(\epsilon^2))}{\epsilon} \\
&= \lim_{\epsilon \to 0^+} \frac{\epsilon a(t+\epsilon)\nabla^2_{\hat{Y}_t} \log p_B(\hat{Y}_t, t) + \mathcal{O}(\epsilon^2)}{2\epsilon} \\
&= \frac{1}{2}a(t)\nabla^2_{\hat{Y}_t} \log p_B(\hat{Y}_t, t),
\end{aligned}
\tag{54}
$$

where $\nabla^2$ is the *Hessian operator*. Based on Lemma B.2.1, theorem 8 and the fact that $a(T) = \nabla_{\hat{Y}_T} \hat{Y}_T = I$, $a(t) = \nabla_{\hat{Y}_t} \hat{Y}_T$ is symmetric. Then theorem 7 shows that the equation $\nabla_{\hat{Y}_t} u(\hat{Y}_t) = \hat{Y}_T(\hat{Y}_t)$ has a solution. Furthermore, since $a(t)$ is a matrix exponential of a symmetric matrix, it must be positive semi-definite. Consequently, the solution $u$ is convex. According to theorem 1, the map $\hat{Y}_T(\hat{Y}_t)$ is optimal for the quadratic transport cost. $\square$

## C.3 Proof of Proposition 3

*Proof.* The definition of $\tilde{Y}_{t+s}$ is given by Equation (23). It can be observed that the term $\tilde{Y}_t(\tilde{Y}_{t+s})$ on the right-hand side indicates that the evolution speed of $\tilde{Y}_{t+s}$ is constant, which implies that all particles travel at a uniform rate. Consequently, for a given initial condition $\tilde{Y}_t$,

$$
\tilde{Y}_{t+s} = \tilde{Y}_t - \frac{1}{2}\nabla_{\tilde{Y}_t} \log p_B(\tilde{Y}_t)s.
\tag{55}
$$

Further, we have:

$$
\nabla_{\tilde{Y}_t} \tilde{Y}_{t+s} = I - \frac{1}{2}\nabla^2_{\tilde{Y}_t} \log p_B(\tilde{Y}_t)s.
\tag{56}
$$

It is evident that $\nabla_{\tilde{Y}_t}\tilde{Y}_{t+s}$ is symmetric and for small values of $s$, it is also positive semi-definite. Based on the same reasoning as the proof of Proposition 2, we can conclude that the map $\tilde{Y}_{t+s}(\tilde{Y}_t)$ is optimal for the quadratic transport cost. $\qquad\square$

## C.4 Proof of Proposition 4

*proof.* As Proposition 1 establishes that $X_t = \phi_t^{-1}Y_{\beta_t}$, the single-time marginal distribution $p_{OU}(x_t, t)$ for the Ornstein-Uhlenbeck process can be expressed as follows:

$$p_{OU}(xt, t) = \frac{1}{N}\sum_i^N (2\pi\beta_t\phi_t^{-2})^{-\frac{n}{2}}\exp(-\frac{|x_t - \phi_t^{-1}x_i|^2}{2\beta_t\phi_t^{-2}}). \tag{57}$$

The probability flow ODE for Ornstein-Uhlenbeck process is:

$$d\hat{X}_t = [-\theta_t\hat{X}_t - \frac{1}{2}\sigma_t^2\nabla_{\hat{X}_t}\log p_{OU}(\hat{X}_t, t)]dt. \tag{58}$$

We start from $\hat{Y}_t$ with the change of variable $Z_t = \phi_t^{-1}\hat{Y}_{\beta_t}$:

$$
\begin{aligned}
\frac{d}{dt}Z_t =& \frac{d\phi_t^{-1}}{dt}\hat{Y}_{\beta_t} + \phi_t^{-1}\frac{d\hat{Y}_{\beta_t}}{d\beta_t}\frac{d\beta_t}{dt} \\
=& -\phi_t^{-1}\theta_t\hat{Y}_{\beta_t} + \phi_t^{-1}(\sigma_t\phi_t)^2[-\frac{1}{2}\nabla_{\hat{Y}_{\beta_t}}p_B(\hat{Y}_t, \beta_t)] \\
=& -\theta_t Z_t - \frac{1}{2}\phi_t\sigma_t^2\frac{\sum_i\exp(-\frac{|\hat{Y}_{\beta_t}-x_i|^2}{2\beta_t})\frac{\hat{Y}_{\beta_t}-x_i}{\beta_t}}{\sum_j\exp(-\frac{|\hat{Y}_{\beta_t}-x_j|^2}{2\beta_t})} \\
=& -\theta_t Z_t - \frac{1}{2}\sigma_t^2\frac{\sum_i\exp(-\frac{|\phi_t^{-1}\hat{Y}_{\beta_t}-\phi_t^{-1}x_i|^2}{2\beta_t\phi_t^{-2}})\frac{\phi_t^{-1}\hat{Y}_t-\phi_t^{-1}x_i}{\beta_t\phi_t^{-2}}}{\sum_j\exp(-\frac{|\phi_t^{-1}\hat{Y}_{\beta_t}-\phi_t^{-1}x_j|^2}{2\beta_t\phi_t^{-2}})} \\
=& -\theta_t Z_t - \frac{1}{2}\sigma_t^2\frac{\sum_i\exp(-\frac{Z_t-\phi_t^{-1}x_i}{2\beta_t\phi_t^{-2}})\frac{Z_t-\phi_t^{-1}x_i}{\beta_t\phi_t^{-2}}}{\sum_j\exp(-\frac{|Z_t-\phi_t^{-1}x_j|^2}{2\beta_t\phi_t^{-2}})} \\
=& -\theta_t Z_t - \frac{1}{2}\sigma_t^2\nabla_{Z_t}\log p_{OU}(Z_t, t).
\end{aligned}
\tag{59}
$$

As $\phi_0 = 1$ and $\beta_0 = 0$, the initial distribution of $X_0$ and $Z_0$ is the same. Consequently, $\hat{X}_t$ and $Z_t$ follow the same ODE with identical initial conditions. Thus, we have $\hat{X}_t = Z_t = \phi_t^{-1}\hat{Y}_{\beta_t}$ and $\nabla_{\hat{X}_t}\hat{X}_{t+s} = \frac{\phi_t}{\phi_{t+s}}\nabla_{\hat{Y}_{\beta_t}}\hat{Y}_{\beta_{t+s}}$, which is symmetric and positive semi-definite by Proposition 2. Therefore, we can conclude that the map $\hat{X}_{t+s}(\hat{X}_t)$ is optimal for the quadratic transport cost. $\qquad\square$

## C.5 Proof of Proposition 5

*Proof.* For the original process (discrete analogue of Brownian motion), the transition rate is:

$$Q_D{}_j^i = \begin{cases} 1, & d_D(i, j) = 1, \\ \sum_{j\in N(i)} -Q_D{}_j^i, & i = j, \\ 0, & others, \end{cases} \tag{60}$$

where $N(i) = \{k : d_D(i, k) = 1\}$. The Kolmogorov forward equation of this process is written as:

$$
\begin{aligned}
\frac{dP_{Di}(t)}{dt} &= \sum_{i'} P_{Di'}(t) Q_{D i}^{i'} \\
&= P_{Di}(t) \times \sum_{i' \in N(i)} -Q_{D i'}^{i} + \sum_{i' \in N(i)} P_{Di'}(t) \times 1 \\
&\quad + \sum_{i' \in \{k:d_D(i,k)>1\}} P_{Di'}(t) \times 0 \\
&= \sum_{i' \in N(i)} (P_{Di'}(t) - P_{Di}(t)).
\end{aligned}
\tag{61}
$$

In contrast, the transition rate of our new process is:

$$
Q_j^i = \begin{cases}
\frac{ReLU(P_{Di}(t) - P_{Dj}(t))}{P_{Di}(t)}, & d_D(i,j) = 1 \\
\sum_{d_D(i,j)=1} -Q_j^i, & i = j \\
0, & others.
\end{cases}
\tag{62}
$$

Our new process can be written as:

$$
\begin{aligned}
\frac{dP_i(t)}{dt} &= \sum_{i'} P_{i'}(t) Q_i^{i'} \\
&= P_i(t) \times \sum_{i' \in N(i)} -Q_{i'}^{i} \\
&\quad + \sum_{i' \in N(i)} P_{i'}(t) \times \frac{ReLU(P_{Di'}(t) - P_{Di}(t))}{P_{Di'}(t)} \\
&\quad + \sum_{i' \in \{k:d_D(i,k)>1\}} P_{i'}(t) \times 0 \\
&= -\sum_{i' \in N(i)} P_i(t) \frac{ReLU(P_{Di}(t) - P_{Di'}(t))}{P_{Di}(t)} \\
&\quad + \sum_{i' \in N(i)} P_{i'}(t) \frac{ReLU(P_{Di'}(t) - P_{Di}(t))}{P_{Di}'(t)}.
\end{aligned}
\tag{63}
$$

Substitute $P = P_D$ in Equation (63), we get the same form in 61, which means $P_D$ also solves the Equation (63). Thus, $P_i(t) = P_{Di}(t), \forall t \geq 0, i \in \{1, 2, \cdots, S\}^K$, according to Picard-Lindelöf theorem. $\qquad\square$

### C.6 Proof of Proposition 6

Let $a = P(t)$, $b = P(t + \varepsilon)$. As our generator only allows flux between adjacent states, we define the transport map $\Pi^* \in \mathbb{R}^{k \times k}$ as:

$$
\Pi^*{}_j^i = \int_t^{t+\epsilon} P_i(t) Q_j^i(t) \, dt,
\tag{64}
$$

which is the probability transported from state $i$ to state $j$ in the time interval $[t, t + \epsilon]$. As the probability $P(t)$ is continuous with respect to time $t$, we choose the $\epsilon$ such that the sign of all the quantities $P_i(t) - P_j(t)$ for $\{i, j \in \{1, 2, \cdots, S\}^K : d_D(i, j) = 1\}$ do not change. Under this assumption, the flux directions do not change at every state.

We claim that $\Pi^*$ solves the optimal transport problem:

$$\min_{\Pi} \sum_{i,j} \Pi_j^i d_D(i,j),$$

$$s.t. \begin{cases} \sum_i \Pi^i = b, \\ \sum_j \Pi_j = a, \\ \Pi \geq 0. \end{cases} \tag{65}$$

*Proof.* The Lagrangian function for this optimization problem is:

$$L(\Pi, \psi, \phi, \lambda) = \sum_{i,j} \Pi_j^i d_D(i,j) + \psi_i(\Pi_j^i - a_i) + \phi_j(\Pi_j^i - b_j) - \lambda_j^i \Pi_j^i, \tag{66}$$

where $\psi_i$, $\phi_j$ and $\lambda_j^i$ are Lagrange multipliers. According to Remark 5 and the fact that this is a linear programming, $\Pi^*$ is optimal if and only if there exists a set of $\phi_i$, $\psi_j$ and $\lambda_j^i$ that satisfy the following equations for $\forall\, i,j$:

$$d_D(i,j) + \psi_i + \phi_j - \lambda_j^i = 0 \tag{67a}$$

$$\lambda_j^i \geq 0 \tag{67b}$$

$$\lambda_j^i \Pi_j^i = 0 \tag{67c}$$

$$\sum_i \Pi_j^i = b_j \tag{67d}$$

$$\sum_j \Pi_j^i = a_i \tag{67e}$$

$$\Pi_j^i \geq 0. \tag{67f}$$

Firstly, we consider the $i,j$ pairs where $d_D(i,j) \leq 1$. In this case $\Pi_j^{*i}$ may $> 0$ (thus $\lambda_j^i = 0$). Besides, the Equation (64) indicates that $\Pi_i^{*i} > 0$, thus we have $\lambda_i^i = 0$. Then the Equation (67a) comes to:

$$\psi_i + \phi_i = 0. \tag{68}$$

According to the construction of our generator $Q$, there is no mutual flux, thus we obtain:

$$\Pi_{i_1,\dots,i_l+1,\dots,i_K}^{i_1,\dots,i_l,\dots,i_K} \neq 0 \text{ or } \Pi_{i_1,\dots,i_l,\dots,i_K}^{i_1,\dots,i_l+1,\dots,i_K} \neq 0,\ \forall\, i. \tag{69}$$

By substituting this result into complementary slackness condition 67c, we have:

$$\lambda_{i_1,\dots,i_l+1,\dots,i_K}^{i_1,\dots,i_l,\dots,i_K,} = 0 \text{ or } \lambda_{i_1,\dots,i_l,\dots,i_K}^{i_1,\dots,i_l+1,\dots,i_K} = 0. \tag{70}$$

Since $d_D([i_1,\dots,i_l,\dots,i_K],[i_1,\dots,i_l+1,\dots,i_K]) = 1$, from Equation (67a), we can obtain:

$$1 + \psi_{i_1,\dots,i_l,\dots,i_K} + \phi_{i_1,\dots,i_l+1,\dots,i_K} = 0 \text{ or } 1 + \psi_{i_1,\dots,i_l+1,\dots,i_K} + \phi_{i_1,\dots,i_l,\dots,i_K} = 0. \tag{71}$$

Solving Equations (68) and (71) simultaneously, we get:

$$\begin{cases} \psi_{i_1,\dots,i_l+1,\dots,i_K} = 1 + \psi_{i_1,\dots,i_l,\dots,i_K} \\ \phi_{i_1,\dots,i_l+1,\dots,i_K} = -1 + \phi_{i_1,\dots,i_l,\dots,i_K} \end{cases} \text{ or } \begin{cases} \psi_{i_1,\dots,i_l+1,\dots,i_K} = -1 + \psi_{i_1,\dots,i_l,\dots,i_K} \\ \phi_{i_1,\dots,i_l+1,\dots,i_K} = 1 + \phi_{i_1,\dots,i_l,\dots,i_K} \end{cases}. \tag{72}$$

Therefore, given $\psi_{0,\dots,0}$, $\psi_{i_1,\dots,i_K}$ and $\phi_{i_1,\dots,i_K}$ can be calculated by:

$$\psi_{i_1,\dots,i_K} = \psi_{0,\dots,0} + m_{0,\dots,0}^{i_1,\dots,i_K} - n_{0,\dots,0}^{i_1,\dots,i_K}, \tag{73}$$

$$\phi_{i_1,\dots,i_K} = -\psi_{i_1,\dots,i_K}, \tag{74}$$

where $m_{0,\dots,0}^{i_1,\dots,i_K} + n_{0,\dots,0}^{i_1,\dots,i_K} = d_D(0,i)$, $m_{0,\dots,0}^{i_1,\dots,i_K} \in \mathbb{N}_0$, and $n_{0,\dots,0}^{i_1,\dots,i_K} \in \mathbb{N}_0$. $\mathbb{N}_0$ represents the set of all non-negative integers. The quantity $m_{0,\dots,0}^{i_1,\dots,i_K}$ is the number where $\Pi_{i_1,\dots,i_l+1,\dots,i_K}^{i_1,\dots,i_l,\dots,i_K} \neq 0$

**Algorithm 1:** Generative Reverse Process with Discrete Probability Flow (DPF)

---

$t \leftarrow T$

$i_t^{1:K} \sim P_T(i_T^{1:K})$

**while** $t > 0$ **do**

    Compute matrix $P_D^\theta = [P_D^\theta[l,j]]_{K \times S}$, where $P_D^\theta[l,j] = P_{D\,i_{t-\tau}^l = j | i_t \setminus i_t^l}^\theta(t), l = 1, \ldots, K,$

       $j = 1, \ldots, S$ with softmax operation on the results of $K \times S$ forward pass of the model

    Encoded the three candidate states (i.e., $i_t^l$, $i_t^l - 1$ and $i_t^l + 1$) for $i_{t-\tau}^l$ with one-hot code:

      $O_{stay} \leftarrow I_{K \times K}[i_t]; O_{sub} \leftarrow I_{K \times K}[i_t - 1]; O_{add} \leftarrow I_{K \times K}[i_t + 1]$

    Fetch the probability $P_D^\theta$ for the above candidate state: $P_{stay} \leftarrow \sum_j (O_{stay} \circ P_D^\theta);$

      $P_{sub} \leftarrow \sum_j (O_{sub} \circ P_D^\theta); P_{add} \leftarrow \sum_j (O_{add} \circ P_D^\theta)$

    $R_{i_{t-\tau}}^{i_t}(t) \leftarrow O_{sub} \circ ReLU(P_{sub}/P_{stay} - 1) + O_{add} \circ ReLU(P_{add}/P_{stay} - 1) - O_{stay} \circ$

      $(ReLU(P_{sub}/P_{stay} - 1) + ReLU(P_{add}/P_{stay} - 1))$ with Equation (30)

    $P^\theta(i_{t-\tau}^l | i_t) \leftarrow \tau R_{i_{t-\tau}}^{i_t}(t) + O_{stay}$ with Equation (31)

    $i_{t-\tau} \leftarrow \text{Categorical}\left(P^\theta(i_{t-\tau}^l | i_t)\right)$

    $t \leftarrow t - \tau$

**end**

---

and $n_{0,\ldots,0}^{i_1,\ldots,i_K}$ is the number where $\Pi_{i_1,\ldots,i_l,\ldots,i_K}^{i_1,\ldots,i_l+1,\ldots,i_K} \neq 0$. Consequently, we find all the Lagrange multipliers for $d_D(i,j) \leq 1$

Then, we consider $i, j$ pairs when $d_D(i,j) > 1$ which indicates $\Pi_j^{*i} = 0$. We use $\psi_i$ and $\phi_j$ in Equation (73) and (74). To satisfy the KKT condition, we only need to verify that there is $\lambda_j^i$ satisfies Equation (67b) and Equation (67a). From Equation (67a), $\lambda_j^i$ can be written as:

$$\lambda_j^i = d_D(i,j) + \psi_i + \phi_j \tag{75}$$

Let $r_l = \min(i_l, j_l)$, it has:

$$d_D(i,j) = d_D(i,r) + d_D(j,r) \tag{76}$$
$$\psi_i = \psi_r + m_i^r - n_i^r \tag{77}$$
$$\phi_j = -\psi_j = -\psi_r - m_j^r + n_j^r \tag{78}$$
$$d_D(i,r) = m_i^r + n_i^r \tag{79}$$
$$d_D(j,r) = m_j^r + n_j^r \tag{80}$$
$$m_i^r, n_i^r, m_j^r, n_j^r \geq 0. \tag{81}$$

Substitute the above results to Equation (75), we have:

$$\lambda_j^i = 2(m_i^r + n_j^r) \geq 0 \tag{82}$$

As a result, we find all the Lagrange multipliers. Since Equations (67d) and (67e) are naturally satisfied by the construction of $\Pi^*$, we conclude that the KKT conditions are met at $\Pi^*$:

① Primal Feasibility: (67d), (67e), (67f)

② Dual Feasibility: (67a), (67b)

③ Complementary slackness: (67c)

According to Remark 5, the KKT conditions indicate $\Pi^*$ is a solution to the optimal transport problem (65). $\qquad\square$

### C.7 Proof of Proposition 7

This is a special case of Proposition 6, where the generator $Q$ remains constant throughout time.

Table 3: Average MMD between different distributions of data.

| State size | 2 | 5 | 10 |
|---|---|---|---|
| Average MMD | 5.336e-3 | 2.201e-2 | 6.531e-3 |

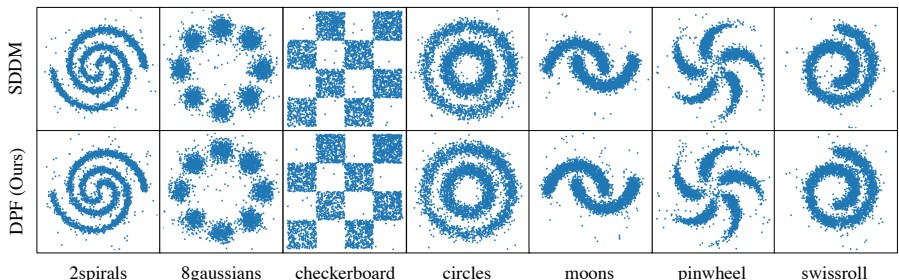

Figure 6: Visualization of the generation quality on generated samples with state size = 5 for SDDM and DPF.

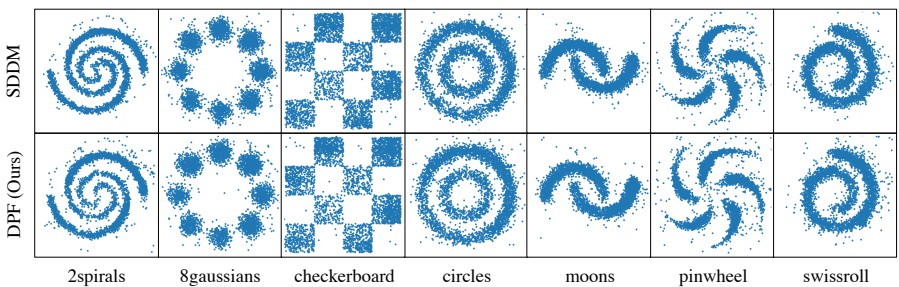

Figure 7: Visualization of the generation quality on generated samples with state size = 10 for SDDM and DPF.

## D    Experiment

### D.1    Algorithm

Our training process follows the same procedure as SDDM, with the distinction that our forward process incorporates the rate we formulated in Equation (28) to align with optimal transport theory. The loss function employed during the training process is as follows:

$$\theta^* = \arg\min_{\theta} \int_0^T \sum_{i_t \in \{1,2,\cdots,S\}^K} q_t(i_t) \left[ \sum_{l=1}^K -\log P_t(i_t^l | i_t \setminus i_t^l) \right] dt. \tag{83}$$

The sampling process with the proposed discrete probability flow is shown in Algorithm 1. In our algorithm, as $R$ is non-zero only when $d_D(i_t, i_{t-\tau}) \leq 1$, the calculation of the reverse transition rate $R$ (as defined in Equation 30) is divided into three cases: staying in the current state ($i_{t-\tau}^l = i_t^l$, i.e., "stay"), jumping to the next state ($i_{t-\tau}^l = i_t^l + 1$, i.e., "add"), and jumping to the previous state ($i_{t-\tau}^l = i_t^l - 1$, i.e., "sub"). By combining the rate in these situations, we can derive $P_\theta(i_{t-\tau}^l | i_t)$ from (31) and (32), which allows us to sample the next state accordingly. This process continues iteratively until $t = 0$.

### D.2    Synthetic Dataset

Following [60, 12, 51], we utilize synthetic data for model validation. Initially, we generate 2D floating-point data from seven distinct distributions using an infinite data oracle. By employing the same settings as [51], we convert each dimension of the data into 16-bit Gray code, resulting in a dataset with discrete dimension = 32 and state size = 2. However, it is not sufficient to validate our method solely on the dataset with state size = 2, since $Q$ in Equation (28) does not cover cases where

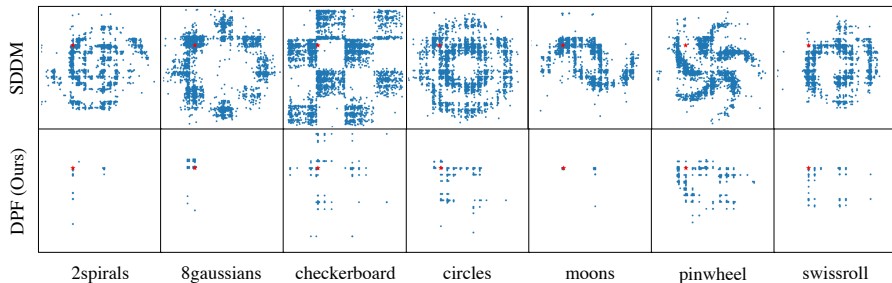

Figure 8: Visualization of the generating certainty on generated samples with state size = 5 for SDDM and DPF. All the samples (in blue) are randomly generated from the single initial point (in red).

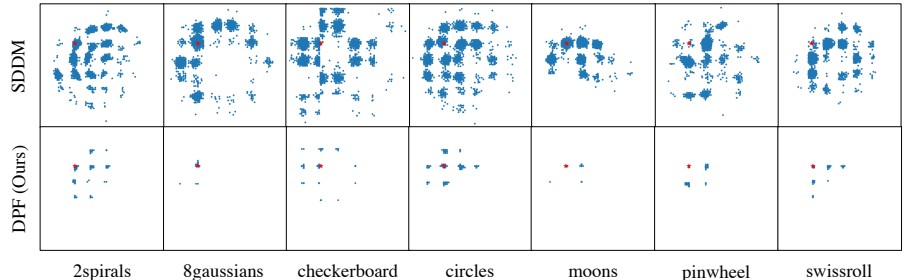

Figure 9: Visualization of the generating certainty on generated samples with state size = 10 for SDDM and DPF. All the samples (in blue) are randomly generated from the single initial point (in red).

$d_D(i, j) > 1$. Therefore, we further transform the same data into 8-bit 5-base code and 6-bit decimal code respectively, thereby creating two additional datasets: one with dimension = 16 and state size = 5, and another with dimension = 12 and state size = 10.

### D.3 Experiment Details

In the experiments, our neural network consists of a 3-layer MLP with 256 channels [60, 51]. We employ the Adam optimizer with a learning rate of 1e-4. The model is trained on a single NVIDIA Quadro RTX 8000, utilizing a batch size of 128 for 300,000 iterations. During training, the parameter $t$ is uniformly sampled from the range of 0 to 1. For the sampling process, the data is generated through 1,000 steps (i.e., $\tau$ is set to 0.001).

### D.4 Quality of Generated Samples

To evaluate the sampling quality, we generate 40,000 samples for binary data, and 4,000 samples for other type of synthetic data by SDDM and our method. Then we compare these generated samples to true data using Laplace MMD. This evaluation is repeated 10 times, and the average results are presented in Table 1. It is worth noting that the unbiased estimation of MMD [17] is an approximation by Monte Carlo method, which may cause negative results. It is observed that MMD score of our method is slightly higher than that of SDDM. This is mainly caused by the approximation of $Q_t$. In the sampling process, two terms are present on the right-hand side of Eq. 10: $\frac{q_t(y)}{q_t(x)}$ and $Q_t(y, x)$. In SDDM, only one term, i.e., $\frac{q_t(y)}{q_t(x)}$ is estimated using a neural network, as $Q_{Dt}(y, x)$ is known. Different from SDDM, both terms in our method are evaluated using quantities approximated by the neural network, since our $Q_t(y, x)$ is dependent on $\frac{q_t(y)}{q_t(x)}$ (Eq. 27). This approximation may lead to slightly inferior quality than the SDDM using precise $Q_{Dt}(y, x)$. Due to this being a neural network fitting error, we currently have no feasible alternative approximations to achieve a superior outcome.

To assess the significance of these differences, we presented the MMD between different distributions of real data in Table 3. Taking this result as a reference, we can find that the gap of the MMD score

Table 4: Comparison of the average $L_1$ distance between the generated samples and initial point. Lower values indicate that the generated sample is closer to initial point.

| | 2spirals | 8gaussians | checkerboard | circles | moons | pinwheel | swissroll |
|---|---|---|---|---|---|---|---|
| | discrete dimension = 32, state size = 2 | | | | | | |
| SDDM | 13.5595 | 13.3025 | 13.4710 | 13.5848 | 13.6485 | 13.4875 | 13.6962 |
| DPF (ours) | 1.5965 | 1.3855 | 0.7525 | 1.1875 | 1.8693 | 1.8135 | 1.6955 |
| | discrete dimension = 16, state size = 5 | | | | | | |
| SDDM | 12.7220 | 12.4698 | 12.4833 | 12.5390 | 12.6745 | 12.6238 | 12.7510 |
| DPF (ours) | 1.5265 | 1.6155 | 0.7090 | 1.1668 | 1.7038 | 1.8088 | 1.5888 |
| | discrete dimension = 12, state size = 10 | | | | | | |
| SDDM | 11.3433 | 11.0083 | 11.0243 | 11.2205 | 11.6850 | 11.3895 | 11.6333 |
| DPF (ours) | 1.7655 | 1.1940 | 0.7143 | 1.1588 | 2.0493 | 1.9283 | 1.7695 |

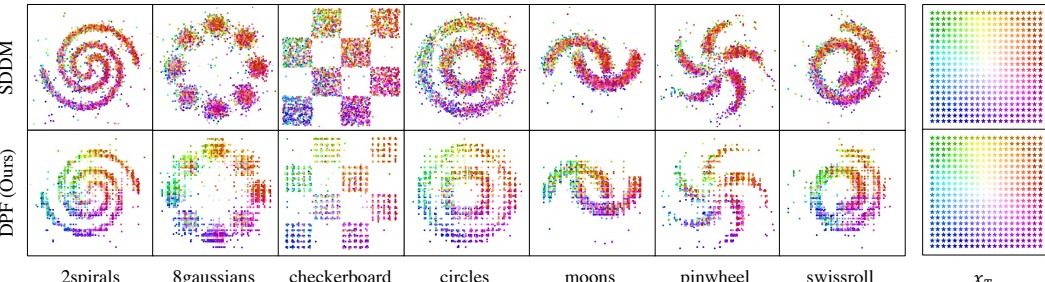

Figure 10: Visualization of the generated samples with state size = 5 from the given initial points $\boldsymbol{x}_T$. Different colors distinguish the generated samples from different initial points $\boldsymbol{x}_T$.

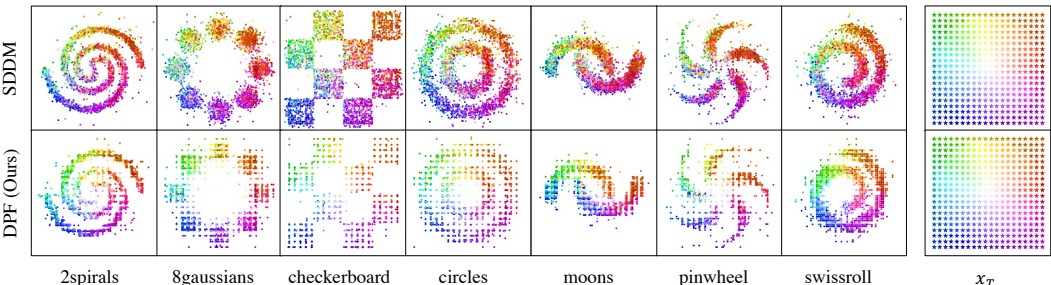

Figure 11: Visualization of the generated samples with state size = 10 from the given initial points $\boldsymbol{x}_T$. Different colors distinguish the generated samples from different initial points $\boldsymbol{x}_T$.

between DPF and SDDM is very small, which is not enough to affect the quality of the generation. This conclusion is further supported by the visualization of the generated samples in Figure 1, Figure 6 and Figure 7 also confirm this point, which demonstrates that DPF produces samples of comparable quality to SDDM.

### D.5 Standard Deviation of Generated Samples

To evaluate the certainty of generated samples, we randomly select a 2D float-point and fix it as the initial point. In this experiments, we use the same initial point (-1.91, 1.57). In the binary case, the point is converted into Gray code, whereas in the 5-base and decimal cases, the original code is utilized. For each dataset, we generate 4,000 points and compute the Expectation of the Conditional Standard Deviation (33). Our DPF method results in a significant reduction in the $CSD$ score, as presented in Table 2. For example, on the checkerboard dataset with $S = 5$, our DPF achieves the best score of 1.4103, which is 89% lower than the score achieved by SDDM. We also visualize these results in Figure 2, Figure 8, and Figure 9, where the red star represents the initial point, and the blue points denote the generated samples. Furthermore, it is evident that SDDM generates samples from a single initial point across the entire space, especially for datasets with $S = 2$. In contrast,

Table 5: Comparison of average trajectory length. Lower value indicates a better transport plan.

| | 2spirals | 8gaussians | checkerboard | circles | moons | pinwheel | swissroll |
|---|---|---|---|---|---|---|---|
| | discrete dimension = 32, state size = 2 | | | | | | |
| SDDM | 32.0075 | 31.7275 | 31.8010 | 32.0258 | 32.0240 | 31.8650 | 31.9988 |
| DPF (ours) | 1.6135 | 1.3980 | 0.7640 | 1.1995 | 1.8883 | 1.8265 | 1.7065 |
| | discrete dimension = 16, state size = 5 | | | | | | |
| SDDM | 26.0680 | 25.3258 | 25.9708 | 25.7565 | 25.8815 | 25.9793 | 26.0810 |
| DPF (ours) | 1.5425 | 1.6390 | 0.7275 | 1.1758 | 1.7102 | 1.8178 | 1.5973 |
| | discrete dimension = 12, state size = 10 | | | | | | |
| SDDM | 21.8558 | 21.7828 | 21.7778 | 21.9100 | 22.3180 | 21.9985 | 22.2688 |
| DPF (ours) | 1.7835 | 1.1995 | 0.7213 | 1.1793 | 2.0623 | 1.9433 | 1.7870 |

our method can only reach a limited number of states from the initial point, indicating the superior sampling certainty of our approach.

We can also observe that our sampling results tend to form rectangles in Figure 2, Figure 8, and Figure 9. This phenomenon arises from the construction of our synthetic dataset. Specifically, we construct the synthetic dataset states and dimensions by encoding the $x$-axis and y-axis coordinates of the toy dataset (normalized to [0, 1]) into $K/2$-bit $S$-ary. This is equivalent to dividing the data space into rectangle regions, where the first few dimensions determine the approximate location of the data. Since our proposed method significantly reduces the uncertainty, each dimension (including the first few dimensions) has only a limited number of possible values. As a result, the points in Figure 2, Figure 8, and Figure 9 appear to form rectangles.

## D.6 Generated Samples from Different Initial Points

To display the generated samples from various initial points, we select a $20 \times 20$ grid of initial points and mark them with distinct colors. Subsequently, we generate 10 samples for each initial point and presented the results in Figure 3, Figure 10, and Figure 11. It is apparent that the samples obtained through SDDM sampling are mixed together. In contrast, the results obtained by our method exhibit strong regularity, with the generated samples clustering together based on their respective colors. This observation suggests that our method offers improved certainty in the sampling process.

## D.7 Distance Between the Generated Samples and Initial Points

Our DPF is designed based on the theory of optimal transport, as demonstrated in Proposition 6. Here, we aim to reflect this finding through experimentation as well. To accomplish this, we utilize the generated samples from Figure 3, Figure 10 and Figure 11, and calculate the average $L_1$ distance from the generated samples to the corresponding initial point:

$$d_D(i(0), i(T)) = \sum_{l=1}^{S} |i^l(0) - i^l(T)|. \tag{84}$$

The results are presented in Table 4. It is evidence that DPF greatly reduces the distance between the generated samples and the initial point. Moreover, combined with the visualization results in Figure 3, Figure 10 and Figure 11, we observe that our method's sampling outcomes tend to concentrate around the high probability states near the initial point. This outcome aligns with our optimal transport design, further verifying the efficacy of our approach.

However, there is an illusion that the difference between SDDM and PDF decreases as the state size increases. This is mainly because that the Figure 3, Figure 10 and Figure 11 are visualized in the 'float space' instead of the 'encoding space'. Specifically, our synthetic data with a state size of and a dimension size of is established by encoding the $x$ and $y$ coordinates of the toy dataset (normalized to [0, 1]) to $K/2$-bit $S$-ary respectively. In this encoding, the first dimension of the encoding has the greatest impact on the data position. For example, in binary encoding (state size = 2), the first bit divides the data space into two parts, and determines the part in which it resides. However, as the number of states increases, the space is divided into more parts, and the small change of the first bit can not significantly change the position of the number it represents. This will lead to a narrowing of the gap between our DPF and SDDM in the visualization. Therefore, in such situations,

Table 6: Comparison of transport efficiency. Larger values indicate better transport efficiency.

| | 2spirals | 8gaussians | checkerboard | circles | moons | pinwheel | swissroll |
|---|---|---|---|---|---|---|---|
| discrete dimension = 32, state size = 2 | | | | | | | |
| SDDM | 42.36% | 41.93% | 42.36% | 42.42% | 42.62% | 42.33% | 42.80% |
| DPF (ours) | 98.95% | 99.11% | 98.49% | 99.00% | 98.99% | 99.29% | 99.36% |
| discrete dimension = 16, state size = 5 | | | | | | | |
| SDDM | 48.80% | 49.24% | 48.07% | 48.68% | 48.97% | 48.59% | 48.89% |
| DPF (ours) | 98.96% | 98.56% | 97.46% | 99.23% | 99.63% | 99.50% | 99.47% |
| discrete dimension = 12, state size = 10 | | | | | | | |
| SDDM | 51.90% | 50.54% | 50.62% | 51.21% | 52.36% | 51.77% | 52.24% |
| DPF (ours) | 98.99% | 99.54% | 99.02% | 98.26% | 99.36% | 99.22% | 99.02% |

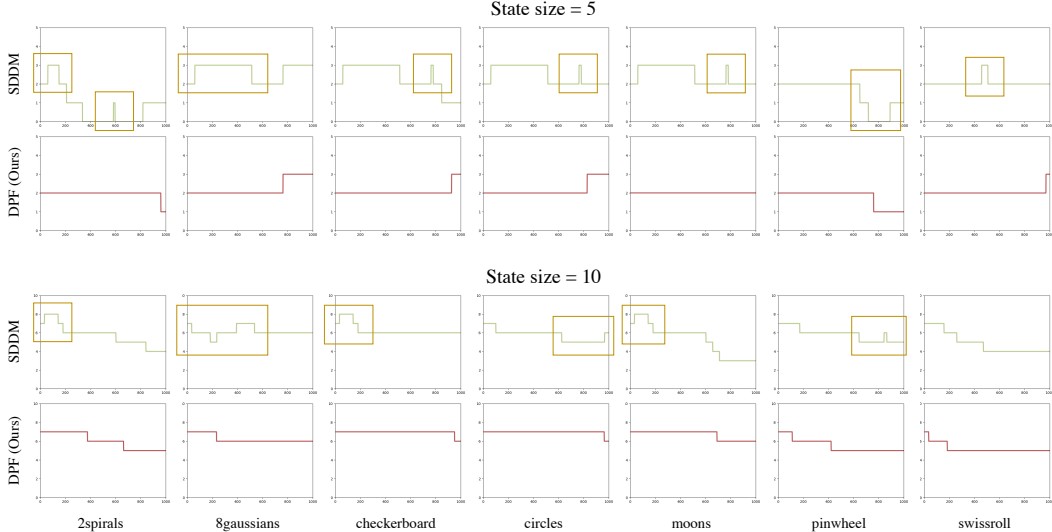

Figure 12: Visualization of the sampling trajectory. The yellow box highlights the duplicated trajectories encountered during the sampling process.

the quantitative results in Table 2 are more appropriate for verifying the reduction of uncertainty in the encoding space.

### D.8    Sampling Trajectory Length

Merely examining the distance between the initial point and the generated samples is insufficient to verify that our DPF aligns with optimal transport principles, as the sampling process may follow different trajectories. Therefore, we calculate the cumulative consumption of the sampling trajectory in Figure 3, Figure 10 and Figure 11 according to the following formula:

$$d_{tra}(i(0), \ldots, i(T)) = \sum_{t \in \{\tau, 2\tau \ldots, T\}} d_D(i(t), i(t - \tau)). \tag{85}$$

The results are shown in Table 5. It is evidence that there is a significant decrease in the trajectory length of DPF compared to the SDDM. For instance, our DPF achieves the best score on the checkerboard dataset with S = 5 with a score of 0.7640, which is 97% lower than the score of SDDM. This suggests that the consumption during our sampling process is lower, which is in line with the optimal transport design.

### D.9    Transport Efficiency

We also examine the transport efficiency during the sampling process, which can be calculated as the ratio of $L_1$ distance between the initial point and generated sample to the sampling trajectory length:

$$E_{(i(0),\ldots,i(T))} = \frac{d_D(i(0), i(T))}{d_{tra}(i(0), \ldots, i(T))}. \tag{86}$$

Table 7: Application on higher dimension or state scenarios. Lower $CSD$ indicate superior certainty.

| | 2spirals | 8gaussians | checkerboard | circles | moons | pinwheel | swissroll |
|---|---|---|---|---|---|---|---|
| | discrete dimension = 20, state size = 50 | | | | | | |
| SDDM | 25.8777 | 26.5288 | 25.5106 | 25.7398 | 25.6984 | 25.6984 | 6.4767 |
| DPF (ours) | 2.7113 | 4.4274 | 2.6217 | 3.7554 | 3.0774 | 3.3054 | 3.8183 |
| | discrete dimension = 50, state size = 5 | | | | | | |
| SDDM | 47.2706 | 47.5810 | 47.4964 | 47.2733 | 47.0047 | 46.9103 | 46.9819 |
| DPF (ours) | 2.0335 | 1.8134 | 0.7418 | 1.7143 | 1.2840 | 1.4245 | 1.5720 |

A higher value indicates a more optimal sampling trajectory selected by the model from the initial point to the generated sample, i.e., the higher transport efficiency. The results are presented in Table 6. Notably, we observed that only approximately 50% of the trajectory length of SDDM contributes to the actual distance between the initial point and generated samples. In contrast, the transport efficiency of our DPF is close to 100%, which means most jumps in our trajectory efficiently contribute to the final transition. This finding demonstrates that the transport plan selected by our DPF is more effective, aligning with our theoretical derivation.

### D.10  Visualization of Sampling Trajectory

The visualization of the sampling trajectory for the 0-th dimension of the dataset is shown in Figure 12. It is evident that the sampling trajectory of SDDM often exhibits duplicate trajectories, which is also the reason for the low transport efficiency of SDDM in Table 6. In contrast, our method, which adheres to the principles of optimal transport theory, ensures that the sampling process only moves toward high probability states, thereby avoiding the occurrence of duplicate trajectories.

### D.11  Higher dimension or state scenarios

To further verify our method is still applicable in higher dimension or state scenarios, we increased the number of states and dimension to 50 for experiments. Specifically, we set $S = 50, K = 20$ and $S = 5, K = 50$ to avoid dimension redundancy in the $K/2$-bit $S$-ary encoding for the toy dataset (float64) coordinates (i.e., $50^{20/2} < 2^{64}$ and $5^{50/2} < 2^{64}$). The results of this experiment are shown in Table 7, which clearly demonstrate that our method can significantly reduce sampling uncertainty even with larger state and dimension sizes.

### D.12  Image Modeling

In addition to the transition rate designed in Eq. 26, our method can also be extended to a broad range of transition rates. For example, we can extend our discrete probability flow to the method in [7]. For general $Q_{D_t}$ with $Q_{D_j^i}(t) = Q_{D_i^j}(t)$, define:

$$Q_j^i(t) = \begin{cases} Q_{D_j^i}(t) \frac{ReLU(P_{D_i}(t) - P_{D_j}(t))}{P_{D_i}(t)}, & i \neq j, \\ -\sum_{j \neq i} Q_j^i, & i = j. \end{cases} \tag{87}$$

$Q_{D_t}$ and $Q_t$ have the same single-time marginal distribution. Let $q_t = P_D(t)$ and $x \neq y$, the reverse transition rate can be written as:

$$R_t(x, y) = \frac{q_t(y)}{q_t(x)} Q_t(y, x) \tag{88a}$$

$$= \frac{q_t(y)}{q_t(x)} Q_{D_t}(y, x) \frac{ReLU(q_t(y) - q_t(x))}{q_t(y)} \tag{88b}$$

$$= Q_{D_t}(y, x) RELU(\frac{q_t(y)}{q_t(x)} - 1) \tag{88c}$$

In the same way, the reverse rate in the paper [7] can be written into the following form:

$$\hat{R}_t^{1:D}(\boldsymbol{x}^{1:D}, \tilde{\boldsymbol{x}}^{1:D}) = \sum_{d=1}^{D} R_t^d(\tilde{x}^d, x^d) \delta_{\boldsymbol{x}^{1:D \setminus d}, \tilde{\boldsymbol{x}}^{1:D \setminus d}} RELU(\sum_{x_0^d} q_{0|t}(x_0^d | \boldsymbol{x}^{1:D}) \frac{q_{t|0}(\tilde{x}^d | x_0^d)}{q_{t|0}(x^d | x_0^d)} - 1)$$
$$\tag{89}$$

Table 8: Comparison of certainty for $\tau$LDR-0 and DPF on the Cifar-10 dataset. Here, $CSD$, class-std, and class-entropy are calculated on 1,000 initial points, each of which has 10 generated images. Lower values indicate superior certainty.

|  | $CSD$ | class-std | class-entropy |
|---|---|---|---|
| $\tau$LDR-0 [7] | 57.6898 | 2.6628 | 1.7703 |
| DPF (ours) | 9.4420 | 1.1819 | 0.5291 |

In this way, the mutual flow between states is eliminated, greatly reducing the sampling uncertainty. To validate this, we validated our DPF on the CIFAR-10 dataset, using the pre-trained discrete diffusion model provided by the paper [7]. Firstly, we selected 1,000 initial points, and sampled 10 images from each initial point. To measure the sampling certainty on the image data, we used a pre-trained CIFAR-10 classifier to classify the image, and introduce two new metrics, i.e., class-std and class-entropy. The class-std calculates the standard deviation of the categories of the images sampled from the same initial point. While the class-entropy calculates the entropy of the category distribution of the images sampled from the same initial point. Lower class-std and class-entropy indicate better sampling certainty. The experimental results, shown in Table 8, demonstrate that our method can significantly reduce the sample uncertainty compared to the $\tau$LDR-0 method. Additionally, we visualized the sampled images in Fig. 4. It was clear that from an initial point, our method samples almost the same images, while the original sampling method obtains totally different images.

## E    Discussion

### E.1    Narrow time interval limited in Proposition 6.

We limit the time frame to a narrow interval, as the validity of the proof hinges on the constancy of the sign of $P_i(t) - P_j(t)$. Alternatively, if both equations in Eq. (70) are established concurrently, a contradiction arises whereby 2 equals 0. Consequently, the KKT condition cannot be satisfied by any suitable Lagrange multipliers, thereby rendering the plan sub-optimal.

From an intuitive standpoint, DPF only avoids instantaneous mutual flow, which does not ensure the elimination of mutual flow during finite interval. For example, if we assume $P_i(t) > P_j(t)$ in the interval $[t, t + \epsilon/2)$ and $P_i(t) < P_j(t)$ in $(t + \epsilon/2, t + \epsilon]$, it follows that $\Pi_j^i > 0$ and $\Pi_i^j > 0$. Assuming $\Pi_j^i > \Pi_i^j$, we can demonstrate that the given plan is sub-optimal. If we define a new plan as $\Pi^{*i}_j = \Pi_j^i - \Pi_i^j$ and $\Pi^{*j}_i = 0$, we can verify that the resultant plan $\Pi^*$ incurs a lower transportation cost than $\Pi$. The preceding derivation establishes the tightness of our announcement, indicating that the optimal transport plan cannot be extended across the entire time interval.

In order to confirm the existence of such a scenario, we explicitly construct it in the case where $K = 1$. Since $P(t) = P(0)e^{Q_D t}$, we can obtain the analytical solution through eigen decomposition with difference equations, yielding the following outcomes: $\lambda_i = 2cos(i\pi/S) - 2$ and $v_i = (1, cos(\theta_i/2), ..., cos((2S - 1)\theta_i/2))$, where $\theta_i = \arccos((\lambda_i + 2)/2)$. Subsequently, we can assess a basic scenario wherein $S = 3$ and $P(t = 0) = (0.1, 0, 0.9)$. It can be observed that $P_0(t = 0) > P_1(t = 0)$ and $P_0(t = 0.1) < P_1(t = 0.1)$. However, the discussion presented above does not deny the existence of a long term optimal transport process. And from an application perspective, it is worth finding out a process with minimal uncertainty.

### E.2    Definition of probability flow on universal discrete process.

In contrast to continuous processes, which necessitate the stochastic term to be a Brownian motion, there are few assumptions regarding the discrete stochastic term. As a result, the consideration of the drift term becomes unnecessary as it can be assimilated into the stochastic term. However, it is worth exploring the potential distinctive properties that may arise from treating these two terms separately.

### E.3    Practical applications.

Analogously to the effect of continuous probability flow on the continuous diffusion model, we believe that reducing uncertainty can also bring many benefits to discrete diffusion models. For

instance, by selecting appropriate initial data, we can generate results that are pertinent to the initial data to attain controllable generation. Additionally, due to the excellent property of sampling certainty reduction, we can perform operations such as interpolation in latent code to complete data editing.

### E.4 Infinite horizon case in Proposition 2.

The infinite horizon case is not addressed in our study due to the presence of singularities that pose significant challenges. For instance, in the case of Brownian motion, the distribution at $t = \infty$ assumes a uniform distribution over the entire $\mathbb{R}^n$, which is not well-defined.

Additionally, the probability flow of Brownian motion at $t = 0$ also experiences a singularity. By taking the limit of the right-hand side of Eq. 20 and let $d_i = x_t - x_i$, we obtain:

$$
\lim_{t \to 0^+} -\frac{1}{2} \nabla_{x_t} p_B(x_t, t) = \lim_{t \to 0^+} \frac{\sum_i exp(-\frac{d_i^2}{2t}) \frac{d_i}{2t}}{\sum_j exp(-\frac{d_j^2}{2t})}
$$

$$
= \lim_{z \to +\infty} \sum_i \frac{d_i}{\sum_j exp((d_i^2 - d_j^2)z)/z}.
$$

Since

$$
\lim_{z \to +\infty} exp((d_i^2 - d_j^2)z)/z = \begin{cases} +\infty & \text{if } d_i^2 > d_j^2, \\ 0 & \text{if } d_i^2 \le d_j^2, \end{cases} \tag{91}
$$

we have

$$
\lim_{t \to 0} -\frac{1}{2} \nabla_{x_t} p_B(x_t, t) = \lim_{z \to +\infty} \frac{d_{i_{min}}}{\sum_j exp((d_{i_{min}}^2 - d_j^2)z)/z}, \tag{92}
$$

where $i_{min} = \arg\min_i d_i^2$. (noting that $i_{min}$ may not be unique, but we exclude this scenario as it does not significantly affect our analysis). Consequently, we obtain:

$$
\lim_{t \to 0^+} -\frac{1}{2} \nabla_{x_t} p_B(x_t, t) = \begin{cases} 0, & \text{if } x_{t=0} = x_{i_{min}}, \\ d_{i_{min}} * \infty, & \text{else.} \end{cases} \tag{93}
$$

where $d_{i_{min}} * \infty$ indicates that the vector is oriented in the direction of $d_{i_{min}}$ and has an infinite norm. Consequently, the right-hand side of Eq. 20 lacks Lipschitz continuity, leading to non-unique solutions. Actually, if Eq. 20 has a unique solution near $t = 0$, the distribution $p_B(x, t)$ will always be a summation of Dirac deltas, which contradicts Eq. 18. Due to our reliance on the solution of ODE, we are unable to analyze the behavior in the vicinity of $t = 0$. Consequently, we have limited our study to finite intervals.