# OpenReview forum: "Formulating Discrete Probability Flow Through Optimal Transport"
_NeurIPS.cc/2023/Conference — NeurIPS 2023 poster_

### Official Review · Reviewer_WcKm · 2023-07-04

**Soundness:** 3 good
**Presentation:** 3 good
**Contribution:** 4 excellent
**Rating:** 7
**Confidence:** 4

**Summary:**

The paper answers two important questions for the diffusion modelling community:

(1) How to interpret the encoding given by the ODE transport map of a diffusion model - it is proven to be an optimal transport map.

(2) How can this ODE type integration be translated to discrete diffusion models - a low stochasticity sampler is derived by again drawing on optimal transport ideas.

The low stochasticity sampler is interpreted on 2d datasets.

**Strengths:**

I like this paper as it very satisfyingly answers the above two important research questions in a very neat way using optimal transport.
Regarding the continuous state space results, the paper has made a significant contribution to the understanding of the 'latent space' induced by a diffusion model integrated using the probability flow ODE. This result has been conjectured before and this paper has now proven the optimal transport interpretation of the flow for a realistic set of assumptions.

Regarding the discrete state space results, the paper has made another significant contribution by formulating what an equivalent 'ODE' integration scheme could be for discrete state spaces, overcoming the challenge imposed by the discrete nature preventing a deterministic map from always being found. Formulating the problem as an optimal transport one seems a very natural way to overcome this. The intuition behind the derived optimal transport reverse rate is also very natural being a bias to only ever move to higher probability states and never flowing backward in such a way as to maintain the same marginals. This is really neat. I went through the proofs for propositions 5-7 and they seem correct to me.

The significance of this paper for the community will be large. For the discrete state spaces I can envision multiple works investigating this new reverse process formulation, for example, given that less transitions occur during sampling, it may be possible to achieve better sampling speed just as ODEs are easier to integrate in continuous state spaces.

**Weaknesses:**

The paper is weak on the experimental side but I believe the strength of the theoretical insights outweigh this.

The paper has not provided any experiments to demonstrate the results on continuous state spaces however I believe these would mostly be similar to Khrulkov et al. The strength of the theoretical results stands alone. Note that I have not the expertise in this area to check the proofs.

The discrete state space experiments are toy but do very clearly present the idea. It is a shame that the authors did not include a practically relevant data type e.g. text as it would have been very interesting to investigate 'discrete interpolations' using this reduced stochasticity sampler. The paper would greatly benefit from an experiment of this type.

The presentation of the paper is poor in places although the overall flow of the paper is good.

In eq(16) you should define what a neighborhood is properly.
On line 178 you should define what you mean by mutual flow.

Typos:

eq (2) dt is in the brackets as well for drift

line 72 x_{t+t} should be x_{t+1}

again line 74 k+t should be k+1

eq 17 extra bracket

eq (26) s=0

line 174 I assume you mean the sum is to K not S?

Edit after rebuttal: I have read the author response and the new experiments alleviated my main concern and I have raised my score to 7.

**Questions:**

The formulation for discrete optimal transport had to include a metric on the discrete state space, which may not be natural for some settings e.g. when the data has no intrinsic ordering. How would you interpret your results in this case, and could the metric perhaps be changed here.

In figures 9 and 10 in the appendix it seems that with bigger state sizes the difference between your low stochasticity sampler and the normal sampler are reduced. Do you expect your results to hold on practical datasets that are quite high dimensional and have a large number of states? Did you test this at all?

**Limitations:**

It is unclear how the method scales to high dimensional / large number of state size datasets.

---

> ### Author Rebuttal · Authors · 2023-08-08
>
> > **The paper has not provided any experiments to demonstrate the results on continuous state spaces however I believe these would mostly be similar to Khrulkov et al.**
>
> Thank you for your comments. In [1], Khrulkov et al. have demonstrated the results on continuous state spaces through numerical experiments, and our contribution is to provide a theoretical basis for their conclusions. More importantly, we focus on analogizing this conclusion to discrete states and designing our discrete probability flows. Therefore, our experiments are primarily conducted around the study of discrete probability flow.
>
> > **The discrete state space experiments are toy but do very clearly present the idea. It is a shame that the authors did not include a practically relevant data type e.g. text as it would have been very interesting to investigate 'discrete interpolations' using this reduced stochasticity sampler. The paper would greatly benefit from an experiment of this type.**
>
> Please refer to our general response.
>
> > **The presentation of the paper is poor in places although the overall flow of the paper is good.**
>
> Thanks for your advice, we will improve our presentation.
>
> > **Definition of "neighborhood" in Eq. (16), and "mutual flow" in line 178.**
>
> Thanks for your advice. We will clarify the two definitions in lines 107 and 178 as follows.
>
>   (1) The discrete model is defined on a graph $G = (V, E)$, where $V = \\{a_1,..., a_N\\}$ is the set of vertices, and $E$ is the set of edges. Therefore, the neighborhood of $i$ in the graph can be defined as:   $N(i) = \\{j \in \\{1, 2,..., N\\}|\\{a_i, a_j\\} \in E\\}$.
>
>  (2) Given two states $i$, $j$, if it has $Q^{i}\_{j} > 0$ and  $Q^{j}\_{i} > 0$, it indicates the presence of mutual flows between states $i$ and $j$.
>
> > **Typos.**
>
> Thanks for your suggestion. We will correct these typo errors.
>
> > **The formulation for discrete optimal transport had to include a metric on the discrete state space, which may not be natural for some settings e.g. when the data has no intrinsic ordering. How would you interpret your results in this case, and could the metric perhaps be changed here.**
>
> Thanks for your comment. This paper aims to propose a discrete probability flow to reduce the sampling uncertainty. Optimal transmission is an approach to studying DPF, which can better explain and characterize the nature and sampling process of DPF, and unify the concept of discrete and continuous probability flows. Therefore, even without the description of the metric, the definition of DPF still applies, and the results can still be explained in terms of reduced sampling uncertainty.
>
> > **In figures 9 and 10 in the appendix it seems that with bigger state sizes the difference between your low stochasticity sampler and the normal sampler are reduced.**
>
> Our method of reducing uncertainty is independent of the number of states and dimensions. However, since Figure 9 and 10 are visualized in the 'float space' instead of the 'encoding space', there is an illusion that the difference between SDDM and PDF decreases as the state increases. Specifically, our synthetic data with a state size of $S$ and a dimension size of $K$ is established by encoding the x and y coordinates of the toy dataset (normalized to [0, 1]) to K/2-bit S-ary respectively. In this encoding, the first dimension of the encoding has the greatest impact on the data position. For example, in binary encoding (state size = 2), the first bit divides the data space into two parts, and determines the part in which it resides. However, as the number of states increases, the space is divided into more parts, and the small change of the first bit can not significantly change the position of the number it represents. This will lead to a narrowing of the gap between our DPF and SDDM in the visualization. Therefore, in such situations, the quantitative results in Table 2 are more appropriate for verifying the reduction of uncertainty in the encoding space.
>
> >**Do you expect your results to hold on practical datasets that are quite high dimensional and have a large number of states? Did you test this at all? Scales to high dimensional / large number of state size datasets?**
>
> Please refer to our general response.
>
> ---
> **Reference:**
>
> [1] Khrulkov, V. et al.. Understanding ddpm latent codes through optimal transport. ICLR, 2023.

---

> > ### Comment · Reviewer_WcKm · 2023-08-11
> >
> > Thank you for the response. I like the new experiment scaling to high dimensionality and showing the minimized variance in sampling effect is still preserved. This has alleviated my main worry and I will increase my score accordingly.

---

> > > ### Author Response · Authors · 2023-08-12
> > > **Thank you very much for the reply**
> > >
> > > We are extremely thankful for your recognition of our work, as well as the time and effort you have dedicated to our paper.

---

### Official Review · Reviewer_v2T5 · 2023-07-05

**Soundness:** 3 good
**Presentation:** 2 fair
**Contribution:** 3 good
**Rating:** 7
**Confidence:** 3

**Summary:**

Building upon the concept of probability flows, the paper establishes a clear link between diffusion processes and optimal transport. It first shows that the continuous probability flow corresponds to a Monge map for every finite-length time interval. It then states and proves analogous statements in the context of continuous-time discrete space (CTDS) models. These results follow the intuition that diffusion should be asymmetric to avoid _mutual flows_ between states. Finally, the authors design a practical sampling method that substantially reduces the variance of end positions of particles, given their initial status.


**Strengths:**

The paper presents a well-executed analysis of diffusion processes from the perspective of probability flows. It modifies a previous result by Khrulkov et al. [29] to make it applicable to the case of Brownian motion with initial distribution given by Eq. 17, i.e., that of a delta train induced by samples.
To analyze the discrete case, the authors revise the (reversed) heat diffusion process by preventing moves to states with lower probability. This procedure, reminiscent of well-known sampling techniques, leads to easy-to-analyze forward (Eq. 27) and backward transition rates. The authors show that the resulting process displaces mass optimally with respect to the L1 cost.


**Weaknesses:**

The notation used throughout the paper is at times hard to parse: Several similar symbols are used concurrently but there is no concise overview of their meaning (and differences). This makes the presentation less intuitive and requires more effort from the reader. In particular, the interpretation of “infinitesimal transport” would benefit from a revision aimed at better disambiguating between processes $\hat{Y}$ and $\tilde{Y}$, and between generators $A$ and $\hat{A}$. Also, the expanded state notation (i.e., $i_1, …, i_K$) is never needed in the main text and could be dropped altogether.
It would also be good to include additional derivations (in the appendix), such as the one of the reverse-time generator $R$ (eq. 29), which are not entirely trivial.

**Experiments**

The authors include several experiments on toy datasets that showcase the effectiveness of the proposed sampling strategy in reducing the variance. They nicely illustrate the theoretical statement which links the process generated by $Q$ to an optimal transport map. However, they do not describe any practical scenarios in which this feature could be necessary or desirable. In particular, reduced variance in sampling seems to imply that slight biases in the initial distribution of particles could lead to unfaithful reconstructions of the other marginal, i.e., by creating “holes” that the optimal transport map fails to fill.
I am also skeptical of the choice of experiments. Why is the current selection restricted to synthetic and low-dimensional settings? Did you run out of time or is the method's applicability limited?

**Miscellaneous**

To ease the description of sampling processes it would be helpful to use the differential (or discrete increment) of time flowing backward, e.g., in Eq. 2.

Further, some **typos**:

- In Table 1, given that the MMD value for the SDDM simulation on the checkerboard dataset has the wrong sign.
- In line 72: $(x_{t+t})$.
- In Eq. 19: missing a ½ factor?
- In the Discussion section.


**Questions:**

What are possible practical applications of the proposed sampling method?

What are possible alternative approximations of the quantity $Q_t$, which is likely the cause of the inferior quality of reconstructed marginals?

Why do points in Figure 2 appear to form rectangles, e.g., in the 2spirals, pinwheel, and swissroll datasets?

Even though the infinite horizon case is outside the scope of the paper, it would be interesting to briefly discuss why the arguments presented in the paper break down in that case. Is this somehow related to mixing times of Markov chains? Additional insight, e.g., in the discussion, could provide a valuable starting point for further research.


**Limitations:**

The practical relevance of the proposed sampling framework is not discussed. It is unclear whether the reduced variance achieved by it could be of use in real-world scenarios and justify the (admittedly) inferior quality of marginal reconstruction quality of DPF.

---

> ### Author Rebuttal · Authors · 2023-08-08
>
> > **Notation is hard to parse ...**
>
> We apologize for not describing the notations. We use $X_t$ and $Y_t$ to represent the OU process and Brownian motion. For consistency, we adopt the notation '^' and '~' to signify the corresponding probability flow and infinitesimal transport. The same convention applies to the generator $A$; however, we apologize for the omission of '^' in Eq. (20) and Eq. (21).
>
> Thanks for your advice. In the revised version, we will use $i$ instead of $(i_1, ..., i_k)$, and include certain non-trivial derivations.
> > **Practical scenarios.**
>
> Please refer to our general response.
> > **Slight biases in the initial distribution ... unfaithful reconstructions of the marginal ...**
>
> When the initial distribution is biased, the marginal distribution will also change, and almost all diffusion models and their sampling methods, such as continuous probability flow, suffer from this problem. Nonetheless, if the initial distribution is free of holes, the marginal distribution will also be free of holes.
> > **Choice of experiments, scalability.**
>
> Please refer to our general response.
> > **Differential of time flowing backward.**
>
> Thank you for your suggestion. We will include the backward differential equation in our paper as follows: $\frac{\partial}{\partial s}q_{s|t}(x_s|x_t)=-\sum_y q_{s|t}(y|x_t)R_t(y,x_s), s<t$.
> > **Typos ... negative MMD ... missing a ½ factor ...**
>
> (1) We use the unbiased estimation $\text{MMD}_u^2$ from Eq.(3) of [1]. As mentioned in [1], it is possible for $\text{MMD}_u^2$ to produce negative values. We will provide additional explanations in our paper.
>
> (2) We appreciate your kind reminder and will correct these equations for the missing factor of ½.
> > **Practical applications.**
>
> Please refer to our general response.
> > **Alternative approximations of $Q\_t$ & inferior quality of reconstructed marginals.**
>
> In the sampling process, two terms are present on the right side of Eq. (9): $\frac{q_t(y)}{q_t(x)}$ and $Q_t(y, x)$. In SDDM, only one term, i.e., $\frac{q_t(y)}{q_t(x)}$ is estimated using a neural network, as ${Q_D}_t(y, x)$ is known. In contrast, our method evaluates both terms using quantities approximated by the neural network, as $Q_t(y, x)$ depends on $\frac{q_t(y)}{q_t(x)}$ (Eq. 27). This approximation may lead to slightly inferior quality than the SDDM using precise ${Q_D}_t(y, x)$. Since this discrepancy arises from the neural network fitting error, we currently have no feasible alternative approximations to achieve a superior outcome.
>
> In order to improve the outcome, it is possible to formulate a discrete counterpart of the technique presented in reference [2], in which the diffusion models are expressed within a variational framework, and the forward process is subject to optimization rather than being predetermined. Through this approach, it may be feasible to approximate the entire right-hand side of Eq.(9), rather than solely $q_t(x)$.
> > **Figure 2 appears to form rectangles.**
>
> This arises from the construction of our synthetic dataset. We construct the synthetic dataset $S$ states and $K$ dimensions by encoding the x-axis and y-axis coordinates of the toy dataset (normalized to [0, 1]) into K/2-bit S-ary. This is equivalent to dividing the data space into $S^{K/2} \times S^{K/2}$ rectangle regions, where the first few dimensions determine the approximate location of the data. Since our proposed method significantly reduces the uncertainty, each dimension (including the first few dimensions) has only a limited number of possible values. As a result, the points in Figure 2 appear to form rectangles.
> > **Infinite horizon case.**
>
> The infinite horizon case is not addressed in our study due to the presence of singularities that pose significant challenges. For instance, in the case of Brownian motion, the distribution at $t=\infty$ assumes a uniform distribution over the entire ${\mathbb{R}^n}$, which is not well-defined.
>
> Additionally, the probability flow of Brownian motion at $t = 0$ also experiences a singularity. By taking the limit of the right-hand side of Eq. (19) and letting $d_i = x_t - x_i$, we obtain:
>
> $$\begin{aligned} \lim_{t \to 0^+} -\frac{1}{2} \nabla_{x_t} p_B (x_t, t) = & \lim_{t \to 0^+} \frac{\sum_i exp(-\frac{d_i^2}{2t}) \frac{d_i}{2t}}{\sum_j exp(-\frac{d_j^2}{2t})} \\\\ = & \lim_{z \to +\infty} \sum_i \frac{d_i}{\sum_j exp((d_i^2 - d_j^2) z) / z}.\end{aligned}$$
> Since
>
> $$\lim_{z \to +\infty} exp((d_i^2 - d_j^2)z) / z =\begin{cases}+\infty, & \text{if}~~d\_i^2 \> d\_j^2,\\\\ 0, & \text{if}~d_i^2 \leq d_j^2, \end{cases}$$
> we have
>
> $$\lim_{t \to 0^+} - \frac{1}{2} \nabla_{x_t} p_B (x_t, t) = \lim_{z \to +\infty} \frac{d_{i_{min}}}{\sum_j exp((d_{i_{min}}^2 - d_j^2) z) / z},$$
> where $i_{min} = \mathop{\arg\min}\limits_{i} d_i^2 $. (noting that $i_{min}$ may not be unique, but we exclude this scenario as it does not significantly affect our analysis). Consequently, we obtain:
>
> $$\lim_{t \to 0^+}-\frac{1}{2} \nabla_{x_t} p_B (x_t, t)=\begin{cases}0, & \text{if}~x_{t=0}=x_{i_{min}},\\\\ d_{i_{min}}*\infty, & \text{else}, \end{cases}$$
> where $d_{i_{min}} * \infty$ indicates that the vector is oriented in the direction of $d_{i_{min}}$ and has an infinite norm. Consequently, the right-hand side of Eq.(19) lacks Lipschitz continuity, leading to non-unique solutions. Actually, if Eq.(19) has a unique solution near $t=0$, the distribution $p_B(x,t)$ will always be a summation of Dirac deltas, which contradicts Eq.(18). Due to our reliance on the solution of ODE, we are unable to analyze the behavior in the vicinity of $t=0$. Consequently, we have limited our study to finite intervals. However, we remain committed to continuing our efforts toward resolving this problem and arriving at a comprehensive conclusion.
>
> ---
> **Reference:**
>
> [1] Gretton, A. et al.. A kernel two-sample test. JMLR, 2012.
>
> [2] Huang C. W. et al.. A variational perspective on diffusion-based generative models and score matching. NeurIPS, 2021.

---

> > ### Comment · Reviewer_v2T5 · 2023-08-10
> > **Thank You for the Rebuttal**
> >
> > I am still at unease about the choice of experiments and feel that at least some empirical evaluation of the method is necessary. The current selection of toy data represents settings where discrete probability flow are not even necessary, and are of a complexity that does not substantiate the paper's claims. I am not sure I understand the point of the argumentation " complete code of SDDM on synthetic data is publicly available for easy reproduction" in the General Response. What speaks against evaluating it at least against some of the "real-world" settings explored in Sun et al. (2023) [47]?

---

> > > ### Author Response · Authors · 2023-08-12
> > > **Thank you very much for the reply**
> > >
> > > Thank you very much for the reply! Since this paper has a sufficient theoretical derivation of the proposed discrete diffusion flow, and its sampling properties have been clearly verified on the synthetic dataset, we apologize for previously neglecting the experiments in the application domain. We greatly appreciate your suggestion and have already started the experiments on the real-world dataset. However, since the code of the real-world dataset needs to be reproduced by ourselves and requires training, it indeed takes some time. We will supplement this experiment in the camera-ready version and discuss the results.

---

> > > ### Author Response · Authors · 2023-08-15
> > > **Thank you for waiting**
> > >
> > > Thank you for waiting. We have validated our DPF on the CIFAR-10 dataset, using the pre-trained discrete diffusion model provided by the paper [1]. Firstly, we selected 1,000 initial points, and sampled 10 images from each initial point. To measure the sampling certainty on the image data, we used a pre-trained CIFAR-10 classifier to classify the image, and introduce two new metrics, i.e., class-std and class-entropy. The class-std calculates the standard deviation of the categories of the images sampled from the same initial point. While the class-entropy calculates the entropy of the category distribution of the images sampled from the same initial point. Lower class-std and class-entropy indicate better sampling certainty. The experimental results, shown in Table 3*, demonstrate that our method can significantly reduce the sample uncertainty compared to the $\tau$LDR-0 method [1]. Additionally, we visualized the sampled images, and it was clear that from an initial point, our method samples almost the same images, while the original sampling method obtains totally different images. Thank you again for your suggestion, we will supplement these results and discuss them in the camera-ready.
> > >
> > > *Table 3\* Comparison of certainty for $\tau$LDR-0 and DPF on the Cifar-10 dataset. Here, $ECV$, class-std, and class-entropy are calculated on 1,000 initial points, each of which has 10 generated images. Lower values indicate superior certainty.*
> > > |  | $ECV$ | class-std | class-entropy|
> > > | :----: | :----: | :----: | :----: |
> > > | **$\tau$LDR-0** [1] | 57.6898 | 2.6628 | 1.7703 |
> > > | **DPF (ours)** | 9.4420 | 1.1819 | 0.5291 |
> > >
> > > ---
> > > **Reference**
> > >
> > > [1] Campbell, A. et al.. A continuous time framework for discrete denoising models. NeurIPS, 2022.

---

> > > > ### Comment · Reviewer_v2T5 · 2023-08-17
> > > > **Updated Score**
> > > >
> > > > Thank you for the continued efforts and running these experiments. I have updated my score accordingly.

---

### Official Review · Reviewer_wEPe · 2023-07-06

**Soundness:** 4 excellent
**Presentation:** 4 excellent
**Contribution:** 4 excellent
**Rating:** 7
**Confidence:** 2

**Summary:**

In this paper, the authors present a theoretical way to define the probability flow for continuous-time diffusion models with discrete state-space, a notion which is largely used for continuous state-space diffusion models, but does not generalize to the discrete setting. They bring several contributions:

- In the continuous setting, they show that the probability flow (under finite time horizon) of the Ornstein-Uhlenbeck process, starting from a finite collection of samples, is the optimal Monge map with respect to the quadratic cost, thus proving the conjecture of [1].

- Then, they extend the notion of probability flow in the discrete setting, when the noising process $P_D$ is chosen as the discrete analogue of the Brownian motion. To do so, they consider a modified version of the transition rate of $P_D$ (with its reverse generator still tractable) and show that it generates a Kantorovitch plan between $P_D(t)$ and $P_D(t+s)$ under a certain cost function, for $t>0$, $0<s<\delta_t$. Relying on the correspondence between optimal transport and probability flow that they highlight in the continuous case, they define the process hence generated as the discrete probability flow of the original process.

- Relying on the score-based framework in continuous time developed by [2], they train two discrete diffusion models, one with the true noising process $P_D$, the other one with the modified noising process generated by $Q$, called Discrete Probability Flow (DPF). They show experimentally that DPF decreases the uncertainty occurring in the sampling phase, without hurting the quality of the generated samples, and give illustration of how DPF empirically satisfies optimal transport in this setting.

[1] Understanding ddpm latent codes through optimal transport, Khrulkov et al., 2022.

[2] Score-based Continuous-time Discrete Diffusion Models, Sun et al., 2022.

**Strengths:**

- Strong theoretical result in the continuous setting. Really good contribution in the field of discrete state space diffusion models, to bridge the gap between diffusion model and optimal transport.

- Paper very well-written, easy to follow.

**Weaknesses:**

The scalability of the experiments.

**Questions:**

In Proposition 6, the existence of a Kantorovitch plan in the discrete setting is not provided for the whole time interval. Do you observe empirically any change in the transport of the particles on the whole time interval, that could  be aligned with this theoretical gap ?

**Limitations:**

The notion of probability flow is only defined for the noising process corresponding to the Brownian motion.

---

> ### Author Rebuttal · Authors · 2023-08-08
>
> > **Scalability of the experiments.**
>
> Please refer to our general response.
>
> > **In Proposition 6, the existence of a Kantorovitch plan in the discrete setting is not provided for the whole time interval. Do you observe empirically any change in the transport of the particles on the whole time interval, that could be aligned with this theoretical gap ?**
>
> We limit the time frame to a narrow interval, as the validity of the proof hinges on the constancy of the sign of $P_i(t) - P_j(t)$.
> Alternatively, if both equations in Eq. (70) are established concurrently, a contradiction arises whereby 2 equals 0.
> Consequently, the KKT condition cannot be satisfied by any suitable Lagrange multipliers, thereby rendering the plan sub-optimal.
>
> From an intuitive standpoint, if we assume $P_i(t) > P_j(t)$ in the interval $[t,t + \epsilon / 2)$ and $P_i(t) < P_j(t)$ in $(t+\epsilon/2, t+\epsilon]$, it follows that $\Pi^i_j > 0$ and $\Pi^j_i > 0$. Assuming $\Pi^i_j > \Pi^j_i$, we can demonstrate that the given plan is sub-optimal. If we define a new plan as ${\Pi^{\*}}^i_j = \Pi^i_j - \Pi^j_i$ and ${\Pi^{\*}}^j_i = 0$, we can verify that the resultant plan ${\Pi^{*}}$ incurs a lower transportation cost than $\Pi$.
> The preceding derivation establishes the tightness of our announcement, indicating that the optimal transport plan cannot be extended across the entire time interval.
>
> In order to confirm the existence of such a scenario, we explicitly construct it in the case where $K=1$.
> Since $P(t) = P(0) e^{Q_{D} t}$, we can obtain the analytical solution through eigen decomposition. The eigen decomposition of $Q_D$ can be obtained through difference equations, yielding the following outcomes: $\lambda_i = 2 cos(i \pi / S) - 2$ and $v_i = (1, cos(\theta_i/2), ... , cos((2S-1)\theta_i/2))$, where $\theta_i = \arccos((\lambda_i + 2) / 2)$.
> Subsequently, we can assess a basic scenario wherein $S = 3$ and $P(t = 0) = (0.1, 0, 0.9)$. It can be observed that $P_{0}(t = 0) > P_{1}(t = 0)$ and $P_{0}(t = 0.1) < P_{1}(t = 0.1)$.
>
> > **The notion of probability flow is only defined for the noising process corresponding to the Brownian motion.**
>
> Thanks for your comment. In contrast to continuous processes, which necessitate the stochastic term to be a Brownian motion, there are few assumptions regarding the discrete stochastic term. As a result, the consideration of the drift term becomes unnecessary as it can be assimilated into the stochastic term. However, it is worth exploring the potential distinctive properties that may arise from treating these two terms separately. We intend to investigate this particular scenario in future research endeavors.

---

> > ### Comment · Reviewer_wEPe · 2023-08-12
> >
> > Thank you for the response. I find the new experiments with higher dimensionality pretty convincing, and I think that it definitely makes the contribution stronger on the experimental side (which was the main weakness for me). Besides this, my theoretical question was well answered and I suggest the authors to include this explanation in the paper. Overall, it is very nice !

---

> > > ### Author Response · Authors · 2023-08-12
> > > **Thank you very much for the reply**
> > >
> > > We are thankful for your appreciation of our work! We will carefully review our paper and incorporate your valuable feedback into the final version of our paper to make it even stronger.

---

### Official Review · Reviewer_Rnpe · 2023-07-07

**Soundness:** 4 excellent
**Presentation:** 4 excellent
**Contribution:** 4 excellent
**Rating:** 8
**Confidence:** 4

**Summary:**

The paper presents a formulation of discrete probability flow using tools of Optimal Transport. First establishing conditions under which continuous probability flows are OT maps, the authors establish a version for discrete case. The authors then define a discrete probability flow using the connection to OT. Experiments compare the proposed method with SDDM showing that the proposed method yields comparable quality with reduced uncertainty.


**Strengths:**

The paper develops discrete probability flows using optimal transport with applications to diffusion modeling. Probability flows, OT, and diffusion models are all topics of great interest in the literature, which is a strength of the paper. I enjoyed the clear exposition of detailed math. The results were also clearly presented an accurately described. Overall, I found the paper to be a quite enjoyable and interesting read.

**Weaknesses:**

I see no major weaknesses of the paper. Always, one can ask for more simulations and comparisons, and that would strengthen the paper. However, the work as presented is already quite strong and interesting, and I do not see further simulations or experiments to be necessary for the author's intended contribution.

Detailed comments:
- Line 102 "This holds under the 2..." It is a bit unclear what is meant. The equation 14 holds? The interpretation from 13->14? Also, under what assumptions?
- Line 105: "using a newly defined metric" what is this metric?
- Line 107, a bit more detail about the neighborhood and eq 16 seems helpful.
- Line 121: "This paper" --> "We" would be clearer. (Also "This paper" in other places.)
- I don't believe SDDM is defined anywhere. It is also a bit surprising to have not discussed in detail SDDM since that is the main comparison.

**Questions:**

None.

---

> ### Author Rebuttal · Authors · 2023-08-08
>
> > **Unclear expression "This holds under the 2..." in Line 102.**
>
> As the gradient flow of the same functional is different under different metrics, we would like to emphasize here that Eq. (13) is the gradient flow of Eq. (14) under the $W_2$ metric. Thanks for your advice. We will refine our expression in our final version.
>
> > **"metric" in Line 105.**
>
> This metric is defined in [1] as Definition 1. Due to its reliance on several definitions and theorems in [1], it is not easy to rephrase it in a concise manner. Thank you for the reminder, we will provide explanations in the footnotes to help readers understand it better.
>
> > **Definition of “neighborhood” in Line 107.**
>
> The discrete model is defined on graph $G = (V, E)$, where $V = \\{ a_1,..., a_N \\}$ is the set of vertices, and $E$ is the set of edges. The $N(i)$ represents the one-ring neighborhood of $i$ in the graph, which can be written as: $N(i) = \\{j \\in \\{1, 2,..., N\\}|\\{a_i, a_j\\} \in E\\}$. We will further supplement this definition in line 107.
>
> > **This paper" --> "We" would be clearer in Line 121.**
>
> We acknowledge the suggestion and will modify the wording from "this paper" to "we".
>
> > **Details of the method SDDM.**
>
> The SDDM (Score-based continuous-time Discrete Diffusion Model) is the method proposed in [2]. We will further supplement this definition in line 199, and provide more details on the SDDM method.
>
> ---
> **Reference:**
>
> [1] Chow S. et al.. Fokker–planck equations for a free energy functional or markov process on a graph. Archive for Rational Mechanics and Analysis, 2012.
>
> [2] Sun, H. et al.. Score-based continuous-time discrete diffusion models. ICLR, 2023.

---

> > ### Comment · Reviewer_Rnpe · 2023-08-10
> > **Re: author response**
> >
> > Thanks for the response and clarifications! I'll highlight that several of my confusions arise from use of the referent "this", which was ambiguous (to me). I would suggest going through the paper and considering whether uses of "this" could be replaced with a more explicit word/phrase.
> >
> > Nice paper, thanks!

---

> > > ### Author Response · Authors · 2023-08-12
> > > **Thank you very much for the reply**
> > >
> > > We are extremely thankful for your appreciation of our work! We greatly appreciate the suggestions in the comments, and we will keep striving to perfect the writing in our paper.

---

### Author Rebuttal · Authors · 2023-08-09

# General Response to All Reviewers

We thank all the reviewers for their time, insightful suggestions, and valuable comments. We are delighted that they find our approach interesting and theoretically sound (**Reviewer Rnpe, wEPe**), establishes a clear link between diffusion processes and optimal transport (**Reviewer v2T5**), and will have large significance for the community (**Reviewer WcKm**). We have also been inspired by the reviewers to further explore the discrete probability flow and attempt to apply the model to various application scenarios in our future study.

First of all, we summarize some common questions and provide our response as follows.

>**Scalability and the selection of synthetic datasets.**

As Proposition 5 established without any assumption of the number of states and dimensions, our method is with good scalability, and guaranteed to scale to any size as well as practical datasets. It is worth noting that any model can use our sampling method as long as its generator follows Eq. (25), including SDDM, as we have the same training procedure. To further verify our method is still applicable in higher dimension or state scenarios, we increased the number of states $S$ and dimension $K$ to 50 for experiments. Specifically, we set $S=50, K=20$ and $S=5, K=50$ to avoid dimension redundancy in the K/2-bit S-ary encoding for the toy dataset (float64) coordinates (i.e., $50^{20/2} < 2^{64}$ and $5^{50/2} < 2^{64}$). The results of this experiment are shown in Table 1\* and Table 2\*, which clearly demonstrate that our method can significantly reduce sampling uncertainty even with larger state and dimension sizes.

In this paper, we focus on establishing the fundamental theory for the probability flow of discrete diffusion models, and verifying its related properties. Currently, we have selected synthetic dataset for validation for two reasons: (1) it allows a more intuitive representation of sampling properties for easy analysis, and (2) complete code of SDDM on synthetic data is publicly available for easy reproduction. Through various metrics and visualizations constructed on the synthetic dataset, the properties of discrete probability flow in terms of optimal transport and uncertainty reduction have been explicitly verified. Thanks for your suggestion, in our future study we will continue to explore this topic and make further attempts in real-world applications.

*Table 1\*. Comparison of certainty for SDDM and DPF on the synthetic dataset with $K = 20$ and  $S = 50$, in terms of $ECV$ on 4,000 initial points, each of which has 10 generated samples. Lower values indicate superior certainty.*
|         | 2spirals | 8gaussians | checkerboard | circles | moons | pinwheel | swissroll |
| :----:| :----: |:----:  | :----: |:----: |:----: |:----: |:----: |
| **SDDM** | 25.8777 | 26.5288 | 25.5106 | 25.7398 | 25.6984 | 25.6984 | 26.4767 |
| **DPF (ours)** | 2.7113 | 4.4274  | 2.6217 | 3.7554 | 3.0774  | 3.3054 | 3.8183



*Table 2\*. Comparison of certainty for SDDM and DPF on the synthetic dataset with $K = 50$ and  $S = 5$, in terms of $ECV$ on 4,000 initial points, each of which has 10 generated samples. Lower values indicate superior certainty.*
|         | 2spirals | 8gaussians | checkerboard | circles | moons | pinwheel | swissroll |
| :----:| :----: |:----:  | :----: |:----: |:----: |:----: |:----: |
| **SDDM** | 47.2706 | 47.5810  | 47.4964 | 47.2733 | 47.0047  | 46.9103 | 46.9819 |
| **DPF (ours)** | 2.0335 | 1.8134  | 0.7418 | 1.7143 | 1.2840 | 1.4245 | 1.5720 |

>**Possible practical applications of the proposed sampling method.**

Thanks for the suggestion, we will supplement the discussion on the practical relevance of the proposed sampling framework. Analogously to the effect of continuous probability flow on the continuous diffusion model, we believe that reducing uncertainty can also bring many benefits to discrete diffusion models. For instance, by selecting appropriate initial data, we can generate results that are pertinent to the initial data to attain controllable generation. Additionally, due to the excellent property of sampling certainty reduction, we can perform operations such as interpolation in latent code to complete data editing.

---

Next, we will provide detailed responses to each reviewer's comments.

---

### Decision · Program_Chairs · 2023-09-21

**Decision:**

Accept (poster)

**Comment:**

This paper presents a theoretical work that defines the probability flow for continuous-time diffusion models with discrete state-space
All reviewers agree that the paper presents some  novelties and a relevant analysis of diffusion processes from the
perspective of OT and probability flow. They all agree that the paper deserves to be published.